# Summertime fluorescence characteristics of atmospheric water-soluble organic carbon in the marine boundary layer of the western Arctic Ocean

Jinyoung Jung[1], Yuzo Miyazaki[2], Jin Hur[3], Yun Kyung Lee[3], Mi Hae Jeon[1], Youngju Lee[1], Kyoung-Ho Cho[1], Hyun Young Chung[1,4], Kitae Kim[1,4], Jung-Ok Choi[1], Catherine Lalande[1], Joo-Hong Kim[1], Taejin Choi[1], Young Jun Yoon[1], Eun Jin Yang[1], and Sung-Ho Kang[1]

[1]Korea Polar Research Institute, 26 Songdomirae-ro, Yeonsu-gu, Incheon 21990, Republic of Korea
[2]Institute of Low Temperature Science, Hokkaido University, Sapporo 060-0819, Japan
[3]Department of Environment & Energy, Sejong University, 209 Neungdong-ro, Gwangjin-gu, Seoul 05006, Republic of Korea
[4]Department of Polar Sciences, University of Science and Technology, Incheon 21990, Republic of Korea

Correspondence to: Jinyoung Jung (jinyoungjung@kopri.re.kr)

**Abstract.** Accelerated warming and a decline in sea ice coverage in the summertime Arctic Ocean can significantly affect the emissions of marine organic aerosols and biogenic volatile organic compounds. However, how these changes affect the characteristics of atmospheric water-soluble organic carbon (WSOC), which plays an important role in the climate system, remains unclear. Thus, to improve our understanding of WSOC characteristics, including its summertime fluorescence characteristics, in the rapidly changing Arctic Ocean, we simultaneously measured atmospheric concentrations of ionic species and WSOC, fluorescence excitation–emission matrix coupled with parallel factor (EEM–PARAFAC) analysis of WSOC, and marine biological parameters in surface seawaters of the western Arctic Ocean during the summer of 2016. WSOC was predominantly present as fine-mode aerosols (diameter < 2.5 μm) (median = 92 %) with the mean concentration being higher in the coastal water areas ($462 \pm 130$ ngC m$^{-3}$) than in the sea ice-covered areas ($242 \pm 88.4$ ngC m$^{-3}$). Moreover, the WSOC in the fine-mode aerosols was positively correlated with the methanesulfonic acid in the fine-mode aerosol samples collected over the sea ice-covered areas (r = 0.88, $p < 0.01$, n = 10), suggesting high sea–air gas exchange and emissions of aerosol precursor gases due to sea ice retreat and increasing available solar radiation during the Arctic summer. Two fluorescent components, humic-like C1 and protein-like C2, were identified by the PARAFAC modeling of fine-mode atmospheric WSOC. The two components varied regionally between coastal and sea ice-covered areas, with low and high fluorescence intensities observed over the coastal areas and sea ice-covered areas, respectively. Further, the humification index of WSOC was correlated with the fluorescence intensity ratio of the humic-like C1/protein-like C2 (r = 0.89, $p < 0.01$) and the WSOC concentration in the fine-mode aerosols (r = 0.66, $p < 0.05$), with the highest values observed in the coastal areas. Additionally, the WSOC concentration in the fine-mode aerosols was positively correlated with the fluorescence intensity ratio of the humic-like C1/protein-like C2 (r = 0.77, $p < 0.01$), but was negatively correlated to the biological index (r = −0.69, $p < 0.01$). Overall, these results suggested that the WSOC in the fine-mode aerosols in the coastal areas showed a higher degree of

polycondensation and higher aromaticity compared that in the sea ice-covered areas, where WSOC in the fine-mode aerosols was associated with relatively new, less oxygenated, and biologically-derived secondary organic components. The findings can improve our understanding of the chemical and biological linkages of WSOC at the ocean–sea ice–atmosphere interface.

## 1 Introduction

Atmospheric marine aerosols significantly influence the Earth's radiative balance, directly by scattering and absorbing solar radiation, and indirectly by acting as cloud condensation nuclei (CCN) (O'Dowd and De Leeuw, 2007; Quinn and Bates, 2011). The composition and sources of marine organic aerosols are particularly important because organic compounds in aerosols affect the water affinity of aerosols (i.e., hydrophilicity or hydrophobicity) depending on their composition and mixing state, thereby altering CCN formation (Kanakidou et al., 2005). Accordingly, numerous studies have investigated the roles of marine organic aerosols in the climate system, specifically focusing on the characterization and quantification of marine organic aerosols (O'Dowd et al., 2004; Cavalli et al., 2004; Ceburnis et al., 2008; Facchini et al., 2008; Hawkins and Russell, 2010; Russell et al., 2010; Gantt et al., 2011; Miyazaki et al., 2011, 2016; Wilson et al., 2015; Dall'Osto et al., 2017; Jung et al., 2020).

Atmospheric water-soluble organic carbon (WSOC) is a major constituent (approximately 30−60%) of organic carbon (OC) in aerosols in the marine boundary layer (O'Dowd et al., 2004; Facchini et al., 2008; Miyazaki et al., 2016; Jung et al., 2020). The hygroscopicity and CCN activity of aerosols depend on the amount and chemical properties of WSOC in aerosols (Saxena et al., 1995; Matsumoto et al., 1997; Ervens et al., 2005). Detailed chemical analysis of atmospheric WSOC revealed that acidic compounds, including monoacids, diacids, and polyacidic compounds, are a major fraction of WSOC. (e.g., Decesari et al., 2001; Cavalli et al., 2004; Sullivan et al., 2004; Psichoudaki and Pandis, 2013; Xie et al., 2016). In particular, polyacidic compounds, which are composed of aromatic compounds that have aliphatic chains with oxygenated functional groups (e.g., carboxyl, hydroxyl, and carbonyl groups), are often referred to as humic-like substances (HULIS) (Decesari et al., 2001; Kiss et al., 2002; Graber and Rudich, 2006; Salma et al., 2013). HULIS are present ubiquitously in aerosol particles from various environments (e.g., urban, rural, forest, and marine) (e.g., Cavalli et al., 2004; Graber and Rudich, 2006; Hoffer et al., 2006; Fu et al., 2015; Chen et al., 2016; Fan et al., 2016; Frka et al., 2018), fog (Krivácsy et al., 2000), rain (Kieber et al., 2006; Yang et al., 2019), and snow (Voisin et al., 2012). In addition, HULIS constitute a significant fraction (25−75%) of aerosol WSOC (Zheng et al., 2013). However, despite a large number of studies on the investigation of individual or classes of compounds, as mentioned above, complete molecular-level chemical characterization of the WSOC remains currently unavailable (Zheng et al., 2013; Fu et al., 2015). Thus, the lack of comprehensive information regarding the chemical composition of atmospheric WSOC hinders a deeper understanding of the role of WSOC in aerosol characteristics.

A fluorescence excitation–emission matrix coupled with parallel factor analysis (EEM–PARAFAC) is commonly used to investigate the optical and structural characteristics of chromophores that are responsible for light absorption and fluorescence by complex organic matter in terrestrial and oceanic systems (Coble, 1996; Coble et al., 1998; Stedmon et al., 2003; Yamashita

et al., 2008). Chromophoric dissolved organic matter (CDOM) in aquatic environments is relatively well characterized by EEM–PARAFAC (e.g., Coble, 2007; Ishii and Boyer, 2012), whereas it has not been extensively used for the analysis of organic matter in atmospheric aerosols (Mladenov et al., 2011). Chromophore components associated with HULIS and protein-like substances have been determined for WSOC in atmospheric aerosols, suggesting the potential of EEM–PARAFAC in atmospheric analysis (Duarte et al., 2004). Moreover, recent studies have demonstrated that EEM–PARAFAC is a useful tool

to reveal the optical properties and chemical structures of atmospheric WSOC, which provide information regarding its origins, chemical reactions, and formation processes (Lee et al., 2013; Fu et al., 2015; Chen et al., 2016; Miyazaki et al., 2018; Tsui and McNeill, 2018; Jung et al., 2020; Tang et al., 2021; Wu et al., 2021). For example, Miyazaki et al. (2018) found that the fluorescence intensity ratios of HULIS and protein-like substances in the sub-micrometer sea spray aerosols were significantly larger than in the bulk surface seawater, indicating that the sea-to-air transfer of marine organic compounds leads to the

preferential formation of HULIS in the sea spray aerosols. Although EEM–PARAFAC provides a better understanding of the WSOC characteristics, data on the fluorescence properties of atmospheric WSOC at high latitudes, especially in the rapidly changing Arctic Ocean, remain sparse.

The Arctic Ocean has been changing remarkably in recent decades. Particularly, the most evident change is the reduction in the extent of the summer sea ice cover that coincides with an intensive loss of multi-year sea ice (Cavalieri and Parkinson,

2012; Perovich et al., 2020), accompanied by accelerated warming (Ballinger et al., 2020) and upward heat flux from the ocean (Shimada et al., 2006). Further, due to the thinning and decrease in the compactness of sea ice (Perovich et al., 2020), the Arctic Ocean is highly responsive to wind stress (Kwok et al., 2013), which consequently affects the sea-to-air flux of various biogenic compounds, including OC. Additionally, the reduced sea ice coverage can significantly increase the seasonal primary production of marine phytoplankton owing to a longer growing season and an expansive open water area (Arrigo and van

Dijken, 2011, 2015; Lewis et al., 2020). As the summertime Arctic aerosols are primarily influenced by local and regional sources, rather than long-range transport from mid-latitude sources (Quinn et al., 2002; Stohl, 2006), these summertime changes in the Arctic Ocean can significantly affect the emissions of marine organic aerosols, biogenic volatile organic compounds (BVOCs), and atmospheric chemistry. However, studies on the fluorescence properties of atmospheric WSOC in the Arctic are limited (Fu et al., 2015; Park et al., 2019a). Therefore, investigating the fluorescence properties of atmospheric

WSOC in the summertime marine Arctic boundary is important to improve our understanding of the chemical and biological linkages of WSOC at the ocean-sea ice-atmosphere interface.

In this study, simultaneous measurements of aerosol chemical composition and fluorescence properties of WSOC, together with a hydrographic survey, were carried out in the western Arctic Ocean during the summer of 2016 to improve our understanding of the characteristics of atmospheric WSOC. Accordingly, this study aimed to (1) investigate the distributions

of ionic species, which could potentially provide useful information for characterizing sources (e.g., sea spray, biogenic, and anthropogenic) and formation mechanisms (i.e., primary and secondary processes) of WSOC, (2) examine the characteristics of atmospheric WSOC using ionic species and hydrographic data, and (3) characterize the quality and possible formation pathways of atmospheric WSOC with fluorescence EEM–PARAFAC. Subsequently, based on the acquired data, the

differences in the WSOC characteristics between coastal and sea ice-covered areas were discussed to fill the data gaps and
improve further modeling and field observations.

## 2 Methods

Aerosol and seawater samples were collected during the ARA07B cruise conducted in the western Arctic Ocean aboard Korean
icebreaker IBR/V *Araon* (Fig. 1). The cruise started from Nome, Alaska, on 5 August 2016, sailed over the Chukchi Sea and
western Arctic Ocean for 18 d, and returned to Barrow, Alaska, on 22 August 2016. For the purpose of regional analysis, the
study area was geographically divided into two regions: the coastal area (aerosol samples AR1−AR3) and the sea ice-covered
area (aerosol samples AR4−AR13).

### 2.1 Aerosol data

#### 2.1.1 Aerosol sample collection

The aerosol samples were collected continuously with a sampling flow rate of 1000 L min$^{-1}$ on pre-combusted (at 550 °C for
6 h) quartz filters (25 × 20 cm; QR-100, Sibata Scientific Technology Ltd., Japan) using two high-volume aerosol samplers
(HV-1000R, Sibata Scientific Technology Ltd.) placed on the upper deck (20 m a.s.l.) of the ship. Particle size selectors for
particulate matter, $PM_{2.5}$ and $PM_{10}$, were installed on the filters of each aerosol sampler to collect fine-mode (diameter (D) <
2.5 μm) and coarse-mode (2.5 μm < D < 10 μm) aerosols, respectively. A wind-sector controller allowed aerosol samples to
be collected only when the relative wind directions were within ±100°, relative to the ship's bow and when the relative wind
speeds were > 1 m s$^{-1}$ (Jung et al., 2020). The total sampling time was 1−2 d, during which the total sampling volume was
580−1800 m$^3$. In total, 13 aerosol samples were collected during the cruise. After sampling, the filters were frozen at −24 °C
before further chemical analysis. Meteorological variables (e.g., wind speed, wind direction, air temperature, relative humidity,
and solar radiation) were also continuously monitored by a weather monitoring system equipped on the research vessel.

#### 2.1.2 Chemical analyses of ionic species and water-soluble organic carbon

Aerosol samples were analyzed for major ionic species using the method described by Jung et al. (2019, 2020). Briefly, the
collected aerosol samples were divided into four equivalent subsamples. Subsequently, one quarter of each sample was
extracted with 50 mL Milli-Q water (> 18 MΩ·cm, Millipore Co.) using an ultrasonic bath for 30 min. The extraction solution
was then filtered through a 13 mm diameter, 0.45 μm pore size membrane filter (polytetrafluoroethylene syringe filter,
Millipore Co.). The resultant filtrate was analyzed for anions ($Cl^-$, methanesulfonic acid (MSA), $NO_3^-$, and $SO_4^{2-}$) and cations
($Na^+$, $NH_4^+$, $K^+$, $Mg^{2+}$, and $Ca^{2+}$) using ion chromatography (ICS-1100, Thermo Scientific Dionex). Anions were analyzed
using an AS19 anion exchange column (Thermo Scientific Dionex). An eluent generator equipped with an EGC-KOH cartridge
was used to produce potassium hydroxide eluent. Cations were separated and quantified using a CS12A cation exchange
column (Thermo Scientific Dionex). A solution of MSA (99.0% purity, Sigma-Aldrich) was used as the eluent for the cations.

Calibrations were conducted using multilevel standard solutions diluted with stock solutions from Thermo Scientific Dionex

(anion P/N 057590 and cation P/N 046070). The instrumental detection limits were: $Cl^-$, 0.05 µg $L^{-1}$; MSA, 0.02 µg $L^{-1}$; $NO_3^-$, 0.02 µg $L^{-1}$; $SO_4^{2-}$, 0.02 µg $L^{-1}$; $Na^+$, 0.02 µg $L^{-1}$; $NH_4^+$, 0.14 µg $L^{-1}$; $K^+$, 0.16 µg $L^{-1}$; $Mg^{2+}$, 0.08 µg $L^{-1}$; and $Ca^{2+}$, 0.20 µg $L^{-1}$. Based on the replicate injections, the analytical precision was estimated to be < 5%. Further, assuming that all sodium ions ($Na^+$) in aerosols were derived from sea salt, the non-sea-salt sulfate (nss-$SO_4^{2-}$) concentration was calculated as the difference between the total $SO_4^{2-}$ concentration and the sodium ($Na^+$) concentration multiplied by 0.2516, which represents

the $SO_4^{2-}/Na^+$ mass ratio in bulk seawater (Millero and Sohn, 1992).

The other one subsample was ultrasonically extracted using the same method for ionic species measurements. The resultant filtrates were analyzed for WSOC using a total organic carbon (TOC) analyzer (model TOC-L, Shimadzu Inc., Japan). Inorganic carbon was removed by acidifying the samples to pH 2 using 2 M HCl and subsequent sparging for 8 min before conducting the WSOC analysis. The instrument was calibrated using a standard solution of potassium hydrogen phthalate

(Nacalai Tesque Inc., Japan), diluted to different concentrations ranging from 0.5 to 5 mgC $L^{-1}$. Milli-Q water and consensus reference material (42−45 µM C for dissolved organic carbon (DOC); deep Florida Strait water obtained from the University of Miami) were measured at every sixth analysis to check the measurement accuracy. Further, the relative standard deviation of the WSOC analysis for the reproducibility test (at least three measurements per sample) was < 3%.

### 2.1.3 Fluorescence measurement of atmospheric WSOC

The other one subsample was ultrasonically extracted using the same measurement method as that used for ionic species. Three-dimensional fluorescence EEMs were measured using a luminescence spectrometer (Hitachi F-7000, Hitachi Inc., Japan) equipped with a light source of 150 W xenon lamp. The wavelength range of the scanning was set at 250−500 nm for excitation (Ex) with a 5 nm step and 280−550 nm for emission (Em) with a 1 nm step (Jung et al., 2020). The slits for both Ex and Em were fixed at 10 nm. The EEMs of each sample were calibrated by subtracting the EEM of Milli-Q water and were normalized

to Raman unit (RU) by integrating the Raman bands from 380 to 420 nm at a 350 nm excitation (Lawaetz and Stedmon, 2009; Stedmon et al., 2003; Chen et al., 2018). Before calibration, inner filter effects were corrected using absorbance spectra of the same sample (McKnight et al., 2001). Absorbance spectra were measured on an ultraviolet-visible spectrophotometer (Shimadzu 1800, Shimadzu Inc., Japan) using a 1 cm quartz cuvette. The sample EEMs were compiled and characterized by PARAFAC using MATLAB 7.0.4 with the DOMFluor toolbox (Stedmon and Bro, 2008). The number of fluorescent

components was determined based on split-half validation and the percentage of the explained variance (99.3%). The loadings in the Ex and Em for each component were matched to the OpenFluor database with more than 93% similarity (Table 1) (Murphy et al., 2014). The humification index (HIX, ratio of emission intensity 435−480 nm/300−345 nm at 255 nm excitation; Zsolnay et al., 1999), biological index (BIX, ratio of emission intensity 380 nm/430 nm at 310 nm excitation; Huguet et al., 2009), and fluorescence index (FI, ratio of emission intensity 450 nm/500 nm at 370 nm excitation; McKnight et al., 2001)

were calculated from the EEMs. These fluorescence indices have been widely applied in studies of aquatic and terrestrial environments because they provide insights into the sources and chemical properties of chromophores (Zsolnay et al., 1999;

McKnight et al., 2001; Huguet et al., 2009). Moreover, previous studies (e.g., Lee et al., 2013; Fu et al., 2015; Chen et al., 2016; Tang et al., 2021; Wu et al., 2021) revealed that the HIX, BIX, and FI can provide useful information on the degree of humification, the chemical structures, and aging processes of atmospheric WSOC, as detailed in Sect. 3.4.

## 2.2 Sampling and analysis of surface seawater

Surface seawater samples were collected at 31 stations in the Chukchi Sea and western Arctic Ocean to investigate the link between atmospheric WSOC and DOC and chlorophyll-a (Chl-a) using a conductivity–temperature–depth and rosette system holding 24 10-L Niskin bottles (SeaBird Electronics, SBE 911 plus) (Fig. 1).

### 2.2.1 Dissolved organic carbon and in situ chlorophyll-a

Seawater samples for DOC measurements were collected in the Niskin bottles using gravity filtration through an inline pre-combusted (550 °C for 6 h) Whatman GF/F filter held in an acid-cleaned (0.1 M HCl) polycarbonate 47 mm filter holder (PP-47, ADVANTEC) (Jung et al., 2020, 2021). The resultant filtrate was collected in an acid-cleaned glass bottle and subsequently, distributed into two pre-combusted 20 mL glass ampoules using a sterilized serological pipette. Each ampoule was sealed using a torch, quick-frozen, and preserved at −24 °C until further laboratory analysis. Similar to WSOC, DOC was measured using a Shimadzu TOC-L analyzer using the same method. Analytical errors based on the standard deviations for replicated measurements (at least three measurements per sample) were within 5% for DOC (Jung et al., 2021).

To analyze Chl-a, seawater samples were filtered through 47 mm GF/F filters, and then extracted with 90% acetone for 24 h. A fluorometer (Trilogy, Turner Designs, USA) was used to measure Chl-a (Lee et al., 2019).

## 2.3 Backward trajectory analysis

Seven-day backward trajectories were calculated for air masses starting from each aerosol sampling station at altitudes of 500, 1000, and 1500 m above ground level. The Hybrid Single-Particle Lagrangian Integrated Trajectories (HY-SPLIT) model (http://www.ready.noaa.gov/HYSPLIT_traj.php) (Stein et al., 2015) with meteorological input from the National Oceanic and Atmospheric Administration (NOAA) Global Data Assimilation System (GDAS) database was used to analyze the backward trajectories.

## 3 Results and discussion

### 3.1 Major ionic species

### 3.1.1 Sodium ($Na^+$)

The concentration of $Na^+$, an indicator of sea salt aerosol, was 98–820 ng m$^{-3}$, with 61% (median value for all data) of $Na^+$ in the coarse-mode aerosols (Fig. 2a). Generally, wind speed significantly influences the effective production flux of sea salt

aerosols in the marine boundary layer (de Leeuw et al., 2011). However, $Na^+$ concentrations were relatively lower than the mean wind speed (samples AR4−AR8) in sea ice-covered areas, suggesting that the emission of sea salt aerosols by local wind was hindered by sea ice coverage due to decreased wind fetch in the sea ice-covered areas (Nilsson et al., 2001; Held et al., 2011). Additionally, several sea fog events occurred (i.e., air temperature dropped to the dew point, and relative humidity was close to 100%) during the sampling period, and the AR1, AR3, AR4, AR6, AR8, and AR9 aerosol samples were largely or slightly affected by these sea fog events (Fig. 3). Previous studies (Jacob et al., 1984; Bergin et al., 1995; Sasakawa et al., 2003; Herckes et al., 2007; Jung et al., 2013, 2019) demonstrated that compared with fine aerosols, coarse aerosols (D > 2.5 µm) predominantly acted as condensation nuclei of sea fog droplets, and that the water vapor condensation on pre-existing particles increases the aerosol size and accelerates their removal from the atmosphere. Similarly, in this study, $Na^+$ concentration in the coarse-mode aerosols was not correlated with mean wind speed (r = 0.35, $p$ > 0.2), whereas $Na^+$ concentration in the fine-mode aerosols was significantly correlated with the mean wind speed (r = 0.68, $p$ < 0.05). The absence of correlation between the $Na^+$ concentration in the coarse-mode aerosols and mean wind speed implied a lower flux of sea salt aerosols relative to the wind speed in sea ice-covered conditions and/or the preferential removal of $Na^+$ in the coarse-mode aerosols by sea fog.

### 3.1.2 Nitrate ($NO_3^-$)

The $NO_3^-$ concentration ranged from 5.3 to 298 ng m$^{-3}$, with an average of 44 ± 84 ng m$^{-3}$ (Fig. 2b). Similar to prior observations (Leck and Persson, 1996; Hara et al., 1999; Beine et al., 2003; Kawamura et al., 2007), $NO_3^-$ was mainly associated with the coarse-mode aerosols (median = 65%) because gaseous nitric acid derived from nitrogen oxides ($NO_x$) emissions was adsorbed on sea-salt aerosols in the marine atmosphere (Andreae and Crutzen, 1997). Among all aerosol samples collected from the coastal areas (AR1−AR3), AR1 exhibited the lowest $NO_3^-$ concentration (42 ng m$^{-3}$), although the sampling was conducted close to Alaska and Russia (Fig. 1). As $NO_3^-$ was mainly present in the coarse-mode aerosols, the lowest $NO_3^-$ concentration in the coastal areas was attributed to efficient scavenging of coarse aerosols by sea fog, as mentioned previously. When air masses originating from the subarctic western North Pacific Ocean moved over Alaska and thereafter reached the sampling locations of AR2 and AR3, the $NO_3^-$ concentrations in the coastal areas sharply increased from 42 ng m$^{-3}$ to 298 ng m$^{-3}$ (Fig. S1). However, the concentrations drastically decreased when the samples were collected from the sea ice-covered areas of the western Arctic Ocean, with values reaching constant values of 10 ± 4.9 ng m$^{-3}$. A similar decreasing trend of $NO_3^-$ concentration with increasing latitude in the Arctic Ocean was also observed by Yu et al. (2020), suggesting that higher $NO_3^-$ concentrations in AR1−AR3 were most likely affected by strong continental sources than those in other samples (Hole et al., 2009). In addition to continental sources, $NO_x$ can be released in the Arctic by the photochemical loss of $NO_3^-$ in the snowpack through the interaction of UV light with snow surfaces. Moreover, the concomitant oxidation of $NO_x$ by bromine oxide could be an important source of $NO_3^-$, as solar radiation increases during spring and summer (Grannas et al., 2007; Morin et al., 2008). Atmospheric circulation during summer is also confined within the Arctic region (Stohl, 2006; Nielsen et al., 2019), and the backward trajectory analyses showed that air masses circulated over the Arctic Ocean when the

sea ice-covered areas were observed (Fig. S1). Therefore, $NO_3^-$ concentrations determined from the sea ice-covered areas were less likely affected by continental sources and were instead derived from local sources.

### 3.1.3 Methanesulfonic acid (MSA)

The concentration of MSA, a tracer for dimethylsulfide (DMS) input to aerosols, was $10-188$ ng m$^{-3}$, with an average of $72 \pm 56$ ng m$^{-3}$ (Fig. 2c). Although the mean MSA concentration was 6.5 times higher than that observed at Barrow in August ($11 \pm 10$ ng m$^{-3}$; Quinn et al., 2002), it was within the MSA concentration ranges previously observed in the Arctic marine boundary layer (e.g., Leck and Persson, 1996; Kerminen and Leck, 2001). MSA was predominantly associated with the fine-mode aerosols (median $= 80$ %). Further, the MSA concentrations were higher in the sea ice-covered areas than in the coastal areas of the southern Chukchi Sea, whereas in situ surface Chl-a concentrations were high in the southern Chukchi Sea (Fig. S2). Temperature-dependent oxidation pathways of atmospheric DMS (i.e., high MSA yield at low temperatures) can possibly explain the high MSA concentrations at high latitudes (Hynes et al., 1986; Ayers et al., 1991; Bates et al., 1992; Jung et al., 2014, 2020). Indeed, a gradual increase in the MSA concentration from AR1 to AR3 during the sampling periods coincided with a decrease in the air temperature (Figs. 2c and 3). Moreover, the release of DMS, which is produced by whether ice algae and phytoplankton or trapped under the ice or in melt ponds during sea ice retreat or melt, can be attributed to high MSA concentrations in the sea ice-covered areas (Levasseur, 2013; Park et al., 2019b).

### 3.1.4 Non-sea-salt sulfate (nss-SO$_4^{2-}$)

The nss-SO$_4^{2-}$ concentration was $178-421$ ng m$^{-3}$, with an average of $285 \pm 71$ ng m$^{-3}$ (Fig. 2d) and approximately 88% (median value) present in the fine-mode aerosols. Unlike $NO_3^-$, nss-SO$_4^{2-}$ did not show a strong latitudinal gradient, most likely due to the influences of both continental (e.g., fossil fuel combustion and sulfide ore smelting) and marine biogenic (i.e., oxidation of atmospheric DMS from oceanic biological processes) sources (Quinn et al., 2002; Ghahremaninezhad et al., 2016; Nielsen et al., 2019). The mean nss-SO$_4^{2-}$ concentration was approximately 3.5 times lower than that measured at Alert during spring ($820-1000$ ng m$^{-3}$, mean $= 990$ ng m$^{-3}$; Narukawa et al., 2008), but was 3.2 times higher than that observed at Barrow in August ($90 \pm 60$ ng m$^{-3}$; Quinn et al., 2002). These results indicated a strong seasonal variation in nss-SO$_4^{2-}$, with minimum and maximum values observed in summer and winter, respectively (e.g., Quinn et al., 2002, 2007; Leaitch et al., 2018) (see Sect. 3.2 for details). Compared with the previous observation at Barrow (Quinn et al., 2002), the mean nss-SO$_4^{2-}$ concentration was higher in this study possibly because of an apparent influence of marine biogenic sources in the western Arctic Ocean during summer when anthropogenic nss-SO$_4^{2-}$ concentrations are expected to be lower (Quinn et al., 2002). This was supported by the correlation observed between nss-SO$_4^{2-}$ and MSA in the fine-mode aerosols ($r = 0.57$, $p < 0.05$, $n = 13$) (Fig. S3a). Moreover, the correlation coefficient increased slightly from 0.57 to 0.65 when only the aerosols collected over the sea ice-covered areas of the western Arctic Ocean (i.e., samples AR4−AR13) were considered ($r = 0.65$, $p < 0.05$, $n = 10$); however, the marginal change in the correlation coefficient was not statistically strong (Fig. S3b). Furthermore, the mean MSA/nss-SO$_4^{2-}$ ratio in the fine-mode aerosols ($0.21 \pm 0.16$) was comparable to the other summertime values measured in the central

Arctic Ocean during IAOE-91 (0.22, Leck and Persson, 1996) and ASCOS (0.25 ± 0.02, Chang et al., 2011). This supported the apparent influence of marine sources and although the influence of continental sources was minimal in the sea ice-covered areas of the western Arctic Ocean during the cruise (Fig. S1), it cannot be excluded.

## 3.2 WSOC in atmospheric aerosols

The total WSOC concentration of atmospheric aerosols (fine and coarse) during the cruise was 141–656 ngC m$^{-3}$, with an
average of 316 ± 141 ngC m$^{-3}$ (Fig. 4a). The WSOC was predominantly found in the fine-mode aerosols, with the mean WSOC percentage in such aerosols being 92 ± 5.0 % (median = 92 %), which was consistent with the findings of previous studies conducted in other oceanic regions (Cavalli et al., 2004; Jung et al., 2020). The mean total WSOC concentrations observed during sea fog events (AR1, AR3, AR4, AR6, AR8, and AR9; 328 ± 112 ngC m$^{-3}$) were comparable to those during non-sea fog events (AR2, AR5, AR7, AR10−AR13; 307 ± 171 ngC m$^{-3}$), reflecting that WSOC was less likely affected by the
preferential scavenging processes of coarse particles by sea fog than Na$^+$ and NO$_3^-$. Although the influence of sea fog on WSOC concentration in aerosols was not particularly remarkable in this study, it is worth mentioning that sea fog could contribute to the formation of atmospheric WSOC, making favorable conditions for secondary processes (e.g., condensation of organic species on pre-existing aerosol particles) (Blando and Turpin, 2000; Kanakidou et al., 2005; Ervens et al., 2011). Further, the mean total WSOC concentration observed in the western Arctic Ocean was approximately 1.8 times lower than
the mean concentration of total OC in the total suspended particulate aerosols (0.56 ± 0.84 µgC m$^{-3}$; range = 0.11−2.93 µgC m$^{-3}$) in the southern Beaufort Sea in the summer of 2009 (3−25 August) (Fu et al., 2013). However, the mean WSOC concentration was 6.6 times higher than the mean value (47.6 ng m$^{-3}$, range = 7.3−185 ng m$^{-3}$) of chemically identified organic compounds (sugar compounds and fatty acids) in the southern Beaufort Sea. This suggested that most marine organic aerosols in the Arctic Ocean were still not identified at a molecular level (Fu et al., 2013). Additionally, the mean WSOC concentration
was 1.7 times higher (0.186 µgC m$^{-3}$, range = 0.041–0.30 µgC m$^{-3}$) than that observed at Alert in the Canadian High Arctic (Fu et al., 2015), but 1.3 times lower than the average springtime total submicron (PM1) organic mass concentration (0.41 ± 0.36 µgC m$^{-3}$) at Barrow (71.5° N, 156.6° W) in 2008 and 2009 (Frossard et al., 2011).

Atmospheric OC and major ionic species in the Arctic region generally show a strong seasonal variation, with maximum values observed in winter and early spring and minimum concentrations observed in summer, whereas reverse trends were observed
for MSA (Sirois and Barrie, 1999; Quinn et al., 2002; Narukawa et al., 2008; Shaw et al., 2010; Fu et al., 2015; Leaitch et al., 2018). This variation can be attributed to the annual cycles of transport pattern, sea ice, temperature, radiation, biological activity, and atmospheric oxidants (Willis et al., 2018 and references therein). Further, the summertime Arctic aerosols were strongly associated with the transport from oceanic regions due to a decreased meridional transport of atmospheric pollutants from the mid-latitudes to the Arctic (Stohl, 2006). Contrastingly, relatively weak pollutant removal through wet deposition
and long-range transport of continental aerosols and their precursors from the mid-latitudes dramatically increase the atmospheric pollutant concentrations in winter and early spring, and this phenomenon is referred to as Arctic Haze (Sirois and Barrie, 1999; Quinn et al., 2002; Stohl, 2006; Quinn et al., 2007, 2009; Leaitch et al., 2013, 2018; Willis et al., 2018). This

suggested that marine sources could further contribute to OC in late spring and summer. Moreover, marine organic matter contributes to OC in summer through BVOC emissions (Leaitch et al., 2018) and direct sea spray emissions (Russell et al.,

2010; Frossard et al., 2011, 2014). As our study was conducted during summer, the observed WSOC concentration would be more influenced by the sea-to-air emission of marine organic matter from the Arctic Ocean. Indeed, the WSOC concentration in the fine-mode aerosols was positively correlated with in situ surface Chl-a (r = 0.58, $p < 0.05$, Fig. 4b) and surface DOC concentrations (r = 0.58, $p < 0.05$, Fig. 4c), indicating that the WSOC concentration was influenced by the sea-to-air emission of marine organic matter produced by biological activities. Although these relationships confirm that the observed WSOC was

associated with locally produced marine organic matter, the WSOC concentration was substantially higher than in the Amundsen Sea in West Antarctica (range = 0.070–0.18 μgC m$^{-3}$, mean = 0.097 ± 0.038 μgC m$^{-3}$; Jung et al., 2020), where massive phytoplankton blooms are present (Arrigo et al., 2012). This suggested that continental sources may contribute to the WSOC concentration in the western Arctic Ocean, as supported by the variations in the $NO_3^-$ concentrations (Fig. 2b). Moreover, previous studies conducted in summer reported that organic aerosols were influenced by both marine and

continental sources in the southeast Beaufort Sea (Fu et al., 2013) and in the central Arctic Ocean (Chang et al., 2011). However, the impact of continental sources on OC remains low during summer (Fu et al., 2013).

### 3.3 WSOC/Na$^+$ ratio and relationship with in situ surface concentration

The OC/Na$^+$ ratio in marine aerosols has commonly been used to investigate the organic fraction in sea spray aerosols. Previous studies demonstrated that compared with seawater, the submicron sea spray aerosols produced by bursting of bubbles are

enriched in OC, with an increasing degree of organic enrichment observed with decreasing particle size (O'Dowd et al., 2004; Keene et al., 2007; Facchini et al., 2008; Russell et al., 2010; Frossard et al., 2014; Quinn et al., 2014). Figure 4a shows that the WSOC/Na$^+$ ratios in the fine-mode aerosols (0.49−5.4) were substantially higher than in the coarse-mode aerosols (0.013−1.9). Particularly, the average WSOC/Na$^+$ ratio in the fine-mode aerosols (2.4 ± 1.5) was approximately 6.7 times higher than that in the coarse-mode aerosols (0.36 ± 0.62), which was consistent with previous results, thus, indicating that

OC is enriched in submicron sea spray aerosols. However, the WSOC/Na$^+$ ratios of the fine-mode aerosols observed in this study were higher than those measured previously in submicron primary marine aerosol particles having OC/Na$^+$ ranging from 0.1 to 2 (Russell et al., 2010; Frossard et al., 2014). Similarly, Miyazaki et al. (2020) observed higher WSOC/Na$^+$ ratios (0.1−3.4) in the submicron particles collected from Oyashio and its coastal region, suggesting that the ratio is generally larger in the biologically productive oceanic region (e.g., coastal region) than in the non-productive region (e.g., open ocean)

(Frossard et al., 2014; Miyazaki et al., 2020). Moreover, WSOC/Na$^+$ ratio and Chl-a concentration can be associated if phytoplankton blooms influenced the WSOC/Na$^+$ ratio in the fine-mode aerosols during the cruise. Indeed, the WSOC/Na$^+$ ratio in the fine-mode aerosols was positively correlated with the surface Chl-a concentrations (r = 0.74, $p < 0.01$, Fig. S4a), although some aerosol samples (i.e., AR4 and AR5) showed high values even when the surface Chl-a concentrations were low. In contrast, Quinn et al. (2014) reported high organic mass fractions of OC in submicron sea spray aerosols collected over both

high- and low-chlorophyll oceanic regions, where Chl-a was not correlated with seawater DOC or organic mass fraction of

OC. They also reported that the oceanic source of OC (such as DOC) enriches OC in submicron sea spray aerosols, which is uncoupled from local biological activity as measured by Chl-a over large oceanic regions. However, surface Chl-a and surface DOC concentrations were correlated (r = 0.67, $p < 0.05$, Fig. S4b) in the western Arctic Ocean. This result was consistent with that of previous studies, which reported that surface DOC concentration was generally associated with Chl-a concentration and primary productivity in our study region, especially in the Chukchi Sea (Davis and Benner, 2005; Shen et al., 2012). Overall, the high WSOC/Na$^+$ ratio in the fine-mode aerosols likely resulted from the sea-to-air transfer of the oceanic source of OC partially along with biological productivity.

In addition to the influence of phytoplankton blooms, WSOC/Na$^+$ ratios higher than the typical range (0.1−2) can also be because of secondary contributions of photochemical products of primary organic aerosols and/or marine BVOC to the observed aerosols (Frossard et al., 2014; Miyazaki et al., 2016). Moreover, previous studies reported enhanced contributions of secondary organic aerosols in the Arctic regions from late spring to summer (Fu et al., 2009, 2013; Kawamura et al., 2012). Thus, we investigated the relationship between WSOC and MSA (a well-known secondary product of DMS of marine algal origin) in the fine-mode aerosols and found that the two components were poorly correlated when all the samples collected during the cruise were considered (r = 0.087, $p > 0.05$, n = 13, Fig. 5). This lack of correlation could be likely due to the difference in source strength between WSOC and MSA, such as contributions of continental sources of WSOC and/or temperature-dependent production of MSA, in the coastal areas close to Alaska and Russia (samples AR1−AR3). However, interestingly, WSOC and MSA in the fine-mode aerosols were positively correlated (r = 0.88, $p < 0.01$, n = 10) in the aerosol samples (AR4−AR13) collected over the sea ice-covered areas (Fig. 5).

Notably, sea ice is inhabited by ice algae and phytoplankton that produces DMS and organic matter, which act as barriers to the sea–air exchange of volatile and semi-volatile organic gases (Levasseur, 2013; Willis et al., 2018; Abbatt et al., 2019; Lannuzel et al., 2020). Recent evidence also shows that organic gases can be released to the atmosphere from snow or sea ice by photochemical processes of organic species derived from biological production, atmospheric deposition, or in situ chemical formation within snow and sea ice (Grannas et al., 2007; McNeill et al., 2012). Thus, sea ice retreat, increasing available solar radiation, and increasing temperatures during the Arctic summer could increase the sea–air gas exchange and emissions of aerosol precursor gases, such as DMS and VOCs, and increase the emission of primary marine aerosols from open water, marginal sea ice zones, and open leads through wind-driven and photochemical processes (Grannas et al., 2007; McNeill et al., 2012; Levasseur, 2013; Mungall et al., 2017; Abbatt et al., 2019). Given the aforementioned possible processes in the sea ice-covered areas in summer along with the appearance of sea fog events (Sect. 3.1.1; Fig. 3), the significant correlation observed between WSOC and MSA in the fine-mode aerosols collected over the sea ice-covered areas was as expected. Thus, the WSOC/Na$^+$ ratio, which was higher than the typical range (0.1−2), and the significant correlation between WSOC and MSA in the fine-mode aerosols suggested a strong association of WSOC with secondary processes, such as the condensation of VOCs and related oxidation products on pre-existing sea spray aerosols and the chemical transformation of primary components in the condensed phase (Rinaldi et al., 2010), at least in the sea ice-covered areas.

## 3.4 Fluorescence properties of water-soluble organic carbon

Two components were identified by the PARAFAC analysis (Fig. 6). Component 1 (C1) shows primary and secondary excitation peaks around 230 nm and 295 nm, respectively, and a single emission peak around 410 nm, which is defined as a mixture of Peaks A and M (terrestrial and marine humic-like fluorescence signals, respectively) (Coble, 1996; Ishii and Boyer, 2012). The spectral features of Peak A are derivatives of terrestrial organic material from river runoff, while components similar to Peak M represent DOM of biological and microbial origin (Coble et al., 1998; Yamashita et al., 2013; Gao and

Guéguen, 2017). The humic-like component, corresponding to C1 in this study, has been previously found in oceanic regions influenced by river runoff (Yamashita et al., 2008, 2011; Fellman et al., 2010). Similarly, other studies have identified this humic-like component in the Arctic Ocean (Gao and Guéguen, 2017; Brogi et al., 2019) due to a strong influence of terrestrial DOC from Arctic rivers (Holmes et al., 2012; Jung et al., 2021). Component 2 (C2) shows a primary excitation maximum at 225 nm, followed by a secondary peak at 270 nm and an emission maximum at 330 nm, which corresponds to Peak T, is

considered as a protein-like (or tryptophan-like) labile component produced from biological production in marine environments (Coble, 1996; Coble et al., 1998; Yamashita and Tanoue, 2008; Dainard et al., 2015; Gonçalves-Araujo et al., 2016). These fluorescent components were generally comparable to those previously observed in the Arctic seawaters (Dainard et al., 2015; Gonçalves-Araujo et al., 2016; Gao and Guéguen, 2017; Osburn et al., 2017; Chen et al., 2018; Brogi et al., 2019; D'Andrilli and McConnell, 2021).

Humic-like substances in aerosols can originate from primary or secondary sources (Graber and Rudich, 2006). For example, biomass burning (Hoffer et al., 2006; Frka et al., 2018; Pani et al., 2022), fossil fuel combustion (Fan et al., 2016; Qin et al., 2018), and bursting of seawater bubbles (Cavalli et al., 2004; Miyazaki et al., 2018) are major primary sources of humic-like substances in aerosols. Secondary sources include multiphase reactions (e.g., oxidation, condensation, oligomerization, and polymerization) of phenolic and carbonyl compounds, as well as other VOCs from biogenic and anthropogenic sources

(Gelencsér et al., 2002, 2003; Graber and Rudich, 2006; Barsotti et al., 2016; Vidović et al., 2018; Wu et al., 2021 and references therein). The uptake of organic species (e.g., dicarbonyls and dicarboxylic acids) to aqueous aerosols via aqueous-phase oligomerization reactions is another source of secondary organic aerosol materials, which exhibit humic-like characteristics (Graber and Rudich, 2006; Nozière et al., 2007). Additionally, the humic-like substances in polar regions can originate from photochemical processes of biogenic organic species in snow and ice, reemission of natural and anthropogenic

aerosols trapped in the snow or previously deposited on sea ice, and in situ chemical formation within snow and sea ice (Cini et al., 1994, 1996; Grannas et al., 2007; Beine et al., 2012; McNeill et al., 2012; Voisin et al., 2012). Protein-like substances in aerosols can originate from primary emissions and secondary processes from both anthropogenic (e.g., biomass burning) and biogenic (e.g., spores, pollen, terrestrial and marine organic compounds) sources (Chen et al., 2016; Dall'Osto et al., 2017; Jung et al., 2020; Wu et al., 2021).

In our study, the fluorescence intensities of humic-like C1 and protein-like C2 differed regionally, with lower values (0.36 ± 0.046 RU for C1 and 0.22 ± 0.028 RU for C2) observed over the coastal areas (samples AR1−AR3) and higher values (0.44 ±

0.045 RU for C1 and 0.48 ± 0.10 RU for C2) observed over the sea ice-covered areas (samples AR4−AR13) (Fig. 7a). Moreover, the mean contribution of the humic-like C1 in the coastal areas was 62 ± 4.6%, which decreased to 48 ± 5.3% in the sea ice-covered areas (Fig. 7b). Contrastingly, the protein-like C2 showed an opposite trend, with higher contributions in the sea ice-covered areas (52 ± 5.3%) and lower contributions in the coastal areas (38 ± 4.6%), despite the mean WSOC concentration in the fine-mode aerosols in the sea ice-covered areas (242 ± 88.4 ngC m$^{-3}$) being lower than in the coastal areas (462 ± 130 ngC m$^{-3}$).

The WSOC concentration in fine-mode aerosols increased with increasing contribution of the humic-like C1 (r = 0.69, $p < 0.01$), but was negatively correlated with the contribution of the protein-like C2 (r = 0.69, $p < 0.01$) (Fig. 7b). Moreover, the WSOC concentration in fine-mode aerosols was positively correlated with the fluorescence intensity ratio of the humic-like C1/protein-like C2 (r = 0.77, $p < 0.01$) (Fig. 7c). Similar results were found at Alert in the Canadian High Arctic during February–June by Fu et al. (2015), who observed a decrease in the contribution of WSOC to OC with increasing ratio of protein-like fluorescence to humic-like fluorescence in the WSOC fraction. Additionally, Fu et al. (2015) reported that the fluorescence intensity ratio of the protein-like to humic-like components showed positive and negative correlations with the contributions of water-insoluble organic carbon (WIOC) and WSOC to OC, respectively, which were attributed to enhanced sea-to-air emissions of primary marine organic aerosols (i.e., WIOC) in spring. Moreover, Miyazaki et al. (2018) reported higher ratios (1.3−2.6) of the humic-like (Peak M)/protein-like (Peak T) fluorescence intensity in sea spray aerosols, with 411−1519 ngC m$^{-3}$ WSOC concentration, than those (0.52−1.5) in surface seawater. They attributed these results to the preferential partitioning of humic-like compounds in the bubble films that produce sea spray aerosols (Burrows et al., 2014; Elliott et al., 2014), preferential formation of humic-like compounds in sea spray aerosols by aggregation of the precursor materials of the humic-like compound (Gelencsér et al., 2003; Graber and Rudich, 2006; Laskin et al., 2015), or rapid production of microbial marine humic-like compounds accompanied by rapid degradation of the protein-like compound by photochemical and/or microbial activity at the sea surface (Yamashita and Tanoue, 2004).

To elucidate the humic-like C1 properties, we investigated the correlations between the humic-like C1 and the FI, BIX, and HIX, which have been previously used as proxies for the relative contributions of organic matter derived from terrestrial or microbial sources in atmospheric aerosols and in terrestrial and oceanic samples (Lee et al., 2013; Fu et al., 2015; Chen et al., 2016; Miyazaki et al., 2018; Jung et al., 2020; Tang et al., 2021; Wu et al., 2021). In general, FI values of 1.4 or less indicate terrestrially derived organics and high aromaticity, whereas values of 1.9 or higher indicate microbial sources and low aromaticity (McKnight et al., 2001). Further, the increase in BIX is associated with an increase in the contribution of organics derived from microbes. High values (> 1) correspond to a predominantly biological or microbial origin of DOM along with the presence of organic matter freshly released into water, whereas low values (< 0.6) correspond to DOM with a relatively smaller contribution from biological materials (Huguet et al., 2009). HIX represents the degree of humification of organic matter (Zsolnay et al., 1999). High HIX values (> 10) correspond to strongly humified or aromatic organics, principally of terrestrial origin, whereas low values (< 4) indicate compounds of autochthonous or microbial origin (Birdwell and Engel, 2010; Fu et al., 2015; Tang et al., 2021). Moreover, high HIX values indicate a higher degree of polycondensation (low H/C

ratio) and aromaticity (Zsolnay et al., 1999; Qin et al., 2018). Furthermore, the humic-like C1 fluorescence intensity was positively correlated with FI (r = 0.77, $p < 0.01$) and BIX (r = 0.61, $p < 0.05$), but was not correlated with HIX (Figs. 7d−7f). Overall, high humic-like C1 fluorescence intensity values were observed over the sea ice-covered areas (samples AR4−AR13) that corresponded to high FI and BIX values. Conversely, humic-like C1 fluorescence intensity values were low in the coastal areas, and were associated with low FI and BIX values. These results suggested that most humic-like C1, especially over the sea ice-covered areas, was associated with marine biological (or microbial) sources via air-sea-ice (or snow) interactions.

Notably, the fluorescence parameters (i.e., FI, BIX, and HIX) are influenced by photochemical reactions during aging and transport; thus, the possible sources of atmospheric WSOC should be interpreted carefully (Lee et al., 2013; Chen et al., 2016; Wu et al., 2021). For example, Chen et al. (2016) found that the FI and BIX were positively correlated with less-oxygenated ions ($C_xH_y^+$ and $C_xH_yO_1^+$) and negatively correlated with highly oxygenated ions ($CO^+$ and $CO_2^+$) in submicron aerosols collected from urban (Nagoya), forest (Kii Peninsula), and marine environments (tropical Eastern Pacific), implying their association with the oxidation degree of organics. Additionally, Lee et al. (2013) reported that the average values of FI ($1.5 \pm 0.4$) and BIX (0.6) of secondary organic aerosols increased upon aging of secondary organic aerosols, suggesting that fluorescent secondary organic aerosols may potentially be mistaken for biological particles through fluorescence-based detection. Wu et al. (2021) reported that the aging process is probably the main factor influencing the HIX values of atmospheric WSOC, with high and low values observed for aged and fresh aerosols, respectively, since the aging process increases the aromaticity of organic aerosols (Lee et al., 2013). Consequently, the relatively low FI and BIX values, and high HIX values observed in the coastal areas may suggest that the humic-like C1 in the coastal areas has relatively high aromaticity and/or more aged organic aerosols. Comparatively, the correlations of FI and BIX with the humic-like C1 fluorescence intensity and the low HIX values over the sea ice-covered areas could be because the humic-like C1 over the sea ice-covered areas is more affected by relatively new, less-oxygenated, and/or secondary processes (e.g., condensation of VOCs and related oxidation products on pre-existing aerosols) rather than the marine biological sources, although the biological effect on the humic-like C1 cannot be ruled out. This claim can be supported by the strong association of WSOC with secondary processes over the sea ice-covered areas (i.e., significant correlation between WSOC and MSA in the fine-mode aerosols; Sect. 3.3; Fig. 5).

Further investigation on the relationships among fluorescence parameters, fluorescence intensity ratio of the humic-like C1/protein-like C2, and the WSOC concentration in fine-mode aerosols showed that the HIX values were significantly correlated with the fluorescence intensity ratio of the humic-like C1/protein-like C2 (r = 0.89, $p < 0.01$) (Fig. 8a) and WSOC concentration in the fine-mode aerosols (r = 0.66, $p < 0.05$) (Fig. 8b). The highest HIX values were observed in aerosol samples collected from the coastal areas (samples AR1 and AR2). Considering the occurrence of sea fog events (Sect. 3.1.1; Fig. 3), the relatively strong influence of anthropogenic sources (Fig. 2b), high WSOC/$Na^+$ ratio (Sect. 3.3; Fig. 4a), and relationships shown in Figs. 7, 8a, and 8b, the WSOC in fine-mode aerosols in the coastal areas exhibits high polycondensation degree (low H/C ratio) and aromaticity (Zsolnay et al., 1999; Qin et al., 2018). Furthermore, the FI value decreased with increasing WSOC concentration in the fine-mode aerosols, but showed no statistically significant correlation (Fig. 8c). However, the WSOC

concentration in fine-mode aerosols was negatively correlated with the BIX value (r = −0.69, $p < 0.01$) (Fig. 8d). This indicated that the WSOC in the fine-mode aerosols was more likely associated with relatively new, less-oxygenated, and biologically-derived secondary organic components in the sea ice-covered areas than in the coastal areas (Lee et al., 2013; Chen et al., 2016).

## 4 Conclusions

In this study, we reported the summertime fluorescence properties and other WSOC characteristics in marine aerosols collected over the western Arctic Ocean in 2016. Atmospheric concentrations of ionic species and WSOC, EEM spectra of WSOC, and marine biological parameters in surface seawater were measured simultaneously. This provided better understanding of the atmospheric WSOC characteristics in the western Arctic Ocean during summer. The WSOC concentration observed in the western Arctic Ocean (range = 141–656 ngC m$^{-3}$, mean = 316 ± 141 ngC m$^{-3}$) was substantially higher than in the Amundsen
Sea in West Antarctica (range = 70–180 ngC m$^{-3}$, mean = 97 ± 38 ngC m$^{-3}$; Jung et al., 2020), where massive phytoplankton blooms are present (Arrigo et al., 2012). However, a clear distinction in $NO_3^-$ concentration between coastal (158 ± 130 ng m$^{-3}$) and sea ice-covered areas (10 ± 4.9 ng m$^{-3}$) and the MSA/nss-$SO_4^{2-}$ ratios in the fine-mode aerosols (0.21 ± 0.16) comparable to the other summertime values measured in the central Arctic Ocean (0.22, Leck and Persson, 1996; 0.25 ± 0.02, Chang et al., 2011) suggested that the influence of continental sources was minimal, especially in the sea ice-covered areas of
the western Arctic Ocean during summer, although it cannot be excluded. In addition, the positive correlations between WSOC concentration, WSOC/Na$^+$ ratio in the fine-mode aerosols, in situ surface Chl-a, and surface DOC concentrations suggested that the WSOC in aerosols was influenced by the sea-to-air transfer of the oceanic source of OC partially along with biological productivity. During the sampling period, six aerosol samples were largely or slightly affected by sea fog events, but no significant differences in mean total WSOC concentrations were observed between sea fog (328 ± 112 ngC m$^{-3}$) and non-sea
fog events periods (307 ± 171 ngC m$^{-3}$). This result reflects that WSOC in aerosols was less likely affected by the preferential scavenging processes of coarse particles by sea fog than Na$^+$ and $NO_3^-$. However, sea fog could contribute to the formation of atmospheric WSOC, making favorable conditions for secondary processes due to high relative humidity conditions, secondary formation by heterogeneous reactions and/or oligomerization through in-cloud processing. Furthermore, the relationship between WSOC and MSA in the fine-mode aerosols revealed a strong connection of WSOC with secondary processes in the
sea ice-covered areas, probably due to greater sea–air gas exchange and DMS and VOC emissions because of sea ice retreat and increasing available solar radiation during the Arctic summer.

The humic-like C1 and protein-like C2 identified by the EEM–PARAFAC analysis of atmospheric WSOC in the fine-mode aerosols showed that the fluorescence intensities differed regionally between coastal (low intensities) and sea ice-covered areas (high intensities). Moreover, the relationships between the humic-like C1 and fluorescence parameters (FI, BIX, and HIX)
suggested that the humic-like C1 over the sea ice-covered areas was more likely affected by relatively fresh and/or secondary processes rather than marine biological sources; however, the biological effect on the humic-like C1 cannot be completely

excluded. Additionally, the relationship of the WSOC concentration in fine-mode aerosols with the fluorescence intensity ratio of the humic-like C1/protein-like C2, HIX, and BIX was studied. The results suggested that the WSOC in the fine-mode aerosols observed in the coastal areas exhibits higher polycondensation degree (low H/C ratio) and aromaticity than in the sea ice-covered areas, where the WSOC in fine-mode aerosols was more associated with relatively new, less-oxygenated, and biologically-derived secondary organic components.

This study enhances our understanding of summertime atmospheric WSOC characteristics over the western Arctic Ocean, although our results were obtained over a short sampling period. In particular, the EEM–PARAFAC and fluorescence parameters (i.e., FI, BIX, and HIX) of WSOC in aerosols provided information on the chemical structures of water-soluble chromophores and their origins and relative freshness, suggesting that this analytical method is useful for elucidating the chemical reactions and transformation processes of atmospheric WSOC.

Recent hydrographic changes associated with changes in seasonal primary production of marine phytoplankton in response to recent sea ice loss and increased wind mixing have been reported in the Arctic Ocean (Ardyna and Arrigo, 2020; Lewis et al., 2020), which can influence on the productions of primary organic aerosols and secondary organic aerosol precursors that impact aerosol composition and size (Lannuzel et al., 2020). These changes will induce alterations in the quantity and quality of atmospheric WSOC in the Arctic Ocean. In addition, a shift in aerosol chemistry toward more oxygenated and smaller molecules has been reported as the available solar radiation increases after polar sunrise (Fu et al., 2009). Further, the dominance of more oxygenated organic species in Arctic spring results in a more hygroscopic organic component compared to that observed during cleaner times (Willis et al., 2018 and references therein). Consequently, the chemical properties of relatively new and less oxygenated WSOC in the sea ice-covered area are expected to change due to photochemical oxidation processes, altering the hygroscopic properties of WSOC in the Arctic Ocean during summer. Further studies are therefore required to more clearly understand the characteristics of atmospheric OC in the rapidly changing Arctic Ocean.

**Data availability**

The data used in this study are available on request to the corresponding author Jinyoung Jung (jinyoungjung@kopri.re.kr).

**Author contributions**

JJ designed the research, conducted the experiments, processed the data, and wrote the paper. YM, JH, YKL, YL, K-HC, KK, CL, J-HK, TC, and YJY contributed to the scientific discussion and paper correction. EJY and S-HK organized the field campaign and contributed to the scientific discussion and paper correction. MHJ and HYC conducted the experiments. J-OC helped in processing the satellite data.

**Competing interests**

The authors declare that they have no conflict of interest.

## Acknowledgements

We thank the captain and crew of the IBR/V *Araon* for their enthusiastic assistance during the ARA07B cruise.

## Financial support

This research was a part of the project titled "Korea-Arctic Ocean Warming and Response of Ecosystem (K-AWARE)" supported by the Korea Institute of Marine Science & Technology (KIMST) Promotion grant funded by the Ministry of Oceans and Fisheries, Korea (KIMST 20210605) and by the Korea Polar Research Institute (KOPRI) grant funded by the Ministry of Oceans and Fisheries (KOPRI PE22010).

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

Table 1. Excitation (Ex) and emission (Em) maxima of the two fluorescent components, their assignments, and the comparison with previous literatures.

| Components | Ex. (nm) | Em. (nm) | Assignments (Labeled by Coble) | Literature Comparison |
|---|---|---|---|---|
| C1 | 230(295) | 410 | Marine Humic-like (Combination of traditionally defined peak A and M) | C4: <260(305)/404 [Chukchi Seawater] (Chen et al., 2018) |
| | | | | C1: 305/410 [Beaufort Seawater] (Gao and Guéguen, 2017) |
| | | | | C4: 295/405 [Svalbard fjord Seawater] (Brogi et al., 2019) |
| | | | | C1: 300/416 [Greenland Ice core] (D'Andrilli and McConnell, 2021) |
| C2 | 225(270) | 330 | Protein-like (Traditionally defined peak T) | C2: 275/338 [Chukchi Seawater] (Chen et al., 2018) |
| | | | | C3: 273/332 [Greenland Seawater] (Goncalves-Araujo et al., 2016) |
| | | | | C4: 275/320 [Beaufort Seawater] (Dainard et al., 2015) |
| | | | | C5: 240(280)/322 [Greenland Lake] (Osburn et al., 2017) |

The comparison is based on the similarity >93% obtained using the OpenFluor database.

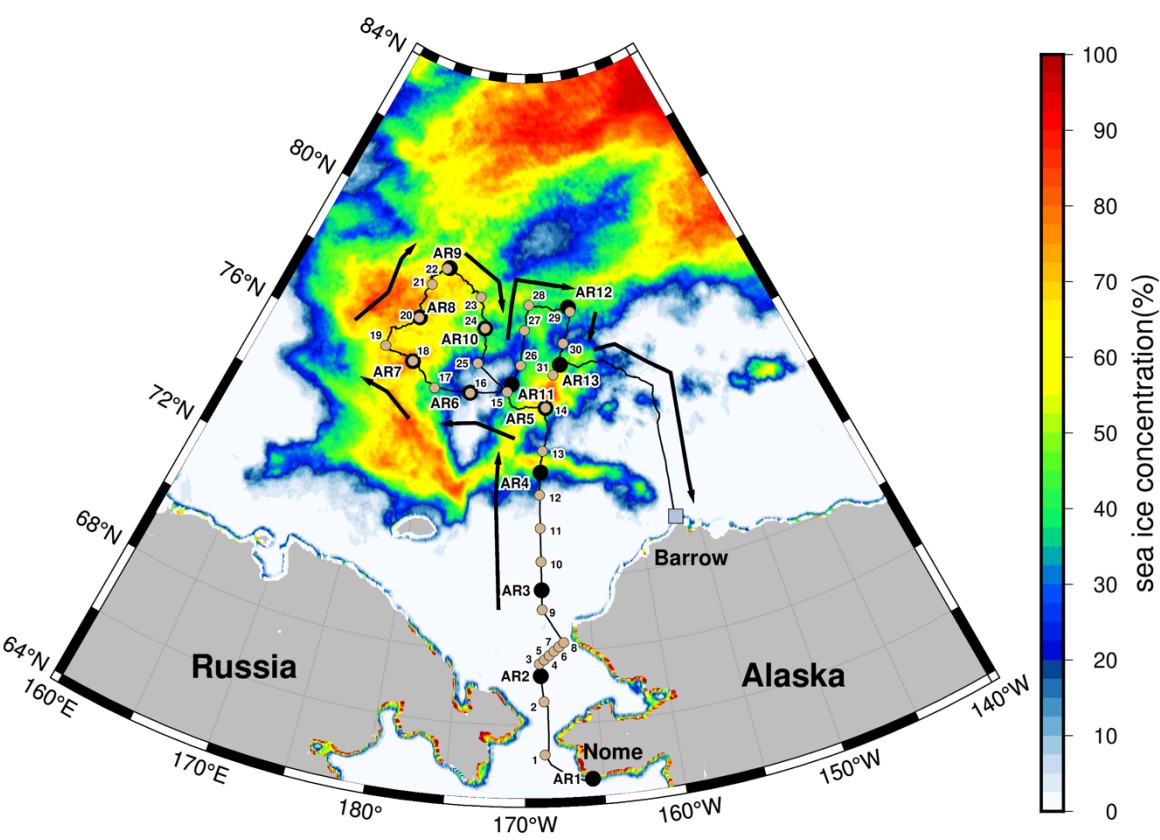

Figure 1: Cruise track (thin black lines) of ARA07B with aerosol (black circles; sample ID: AR1−AR13) and seawater sampling (light brown circles; sample ID: 1−31) locations. Sampling stations were superimposed onto the mean sea ice concentration derived from Advanced Microwave Scanning Radiometer (AMSR) 2 data for August 2016 (Spreen et al., 2008). Each aerosol sampling start point represents the end of the previous sampling period. Black thick arrows indicate the moving direction of the ship.

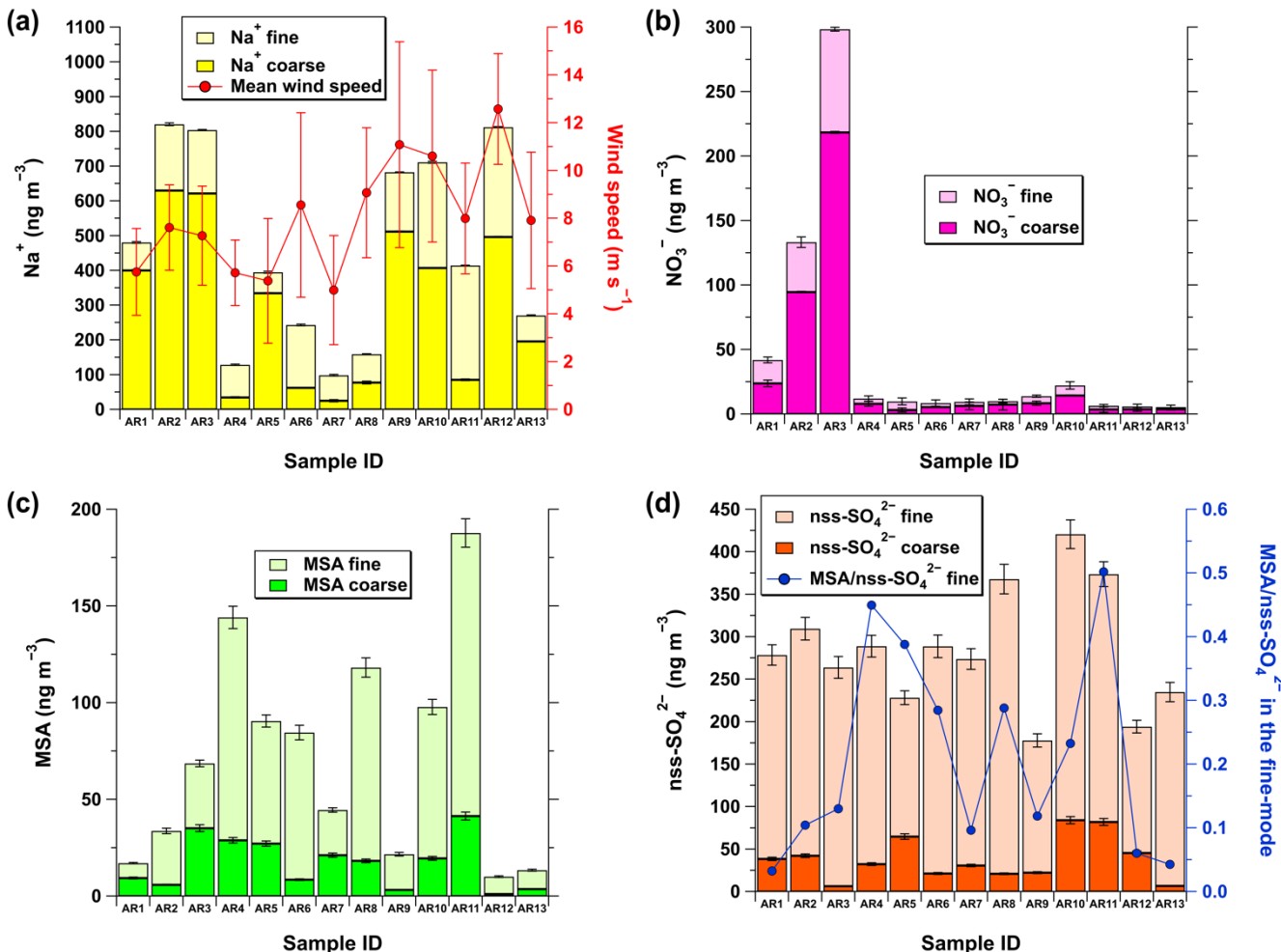

**Figure 2: Concentrations of (a) Na$^+$ (ng m$^{-3}$), (b) NO$_3^-$ (ng m$^{-3}$), (c) MSA (ng m$^{-3}$), and (d) nss-SO$_4^{2-}$ (ng m$^{-3}$) in aerosols collected from the coastal (AR1–AR3) and sea ice-covered areas (AR4–AR13) of the western Arctic Ocean during the summer of 2016. The solid red line with circles and error bars in (a) indicates the mean and standard deviation of wind speeds for each aerosol sampling time. The solid blue line with circles in (d) represents the MSA to nss-SO$_4^{2-}$ ratio in the fine-mode aerosols.**

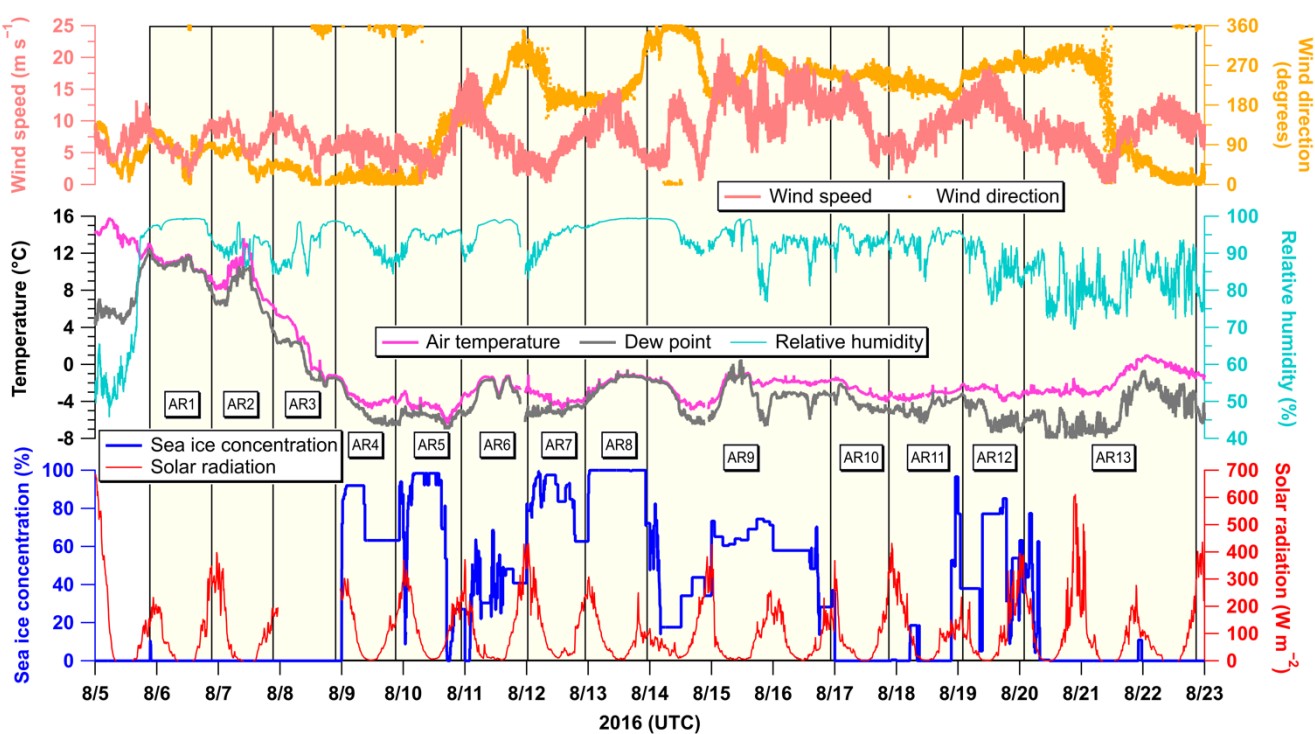

**Figure 3:** Temporal variations of meteorological variables, including wind speed (m s$^{-1}$), wind direction (degrees), air temperature (°C), dew point (°C), relative humidity (%), solar radiation (W m$^{-2}$), and sea ice concentration (%) along the ship's track during the cruise. The yellow hatched area indicates the sampling duration of each aerosol sample (AR1−AR13). Solar radiation was not recorded from 23:20 on 7 August to 23:10 on 8 August due to equipment malfunction. Sea ice concentration represents the percentage of an area covered with sea ice in the ship's location within a 3.125 km × 3.125 km grid derived from AMSR 2 sea ice concentration daily data (Spreen et al., 2008).

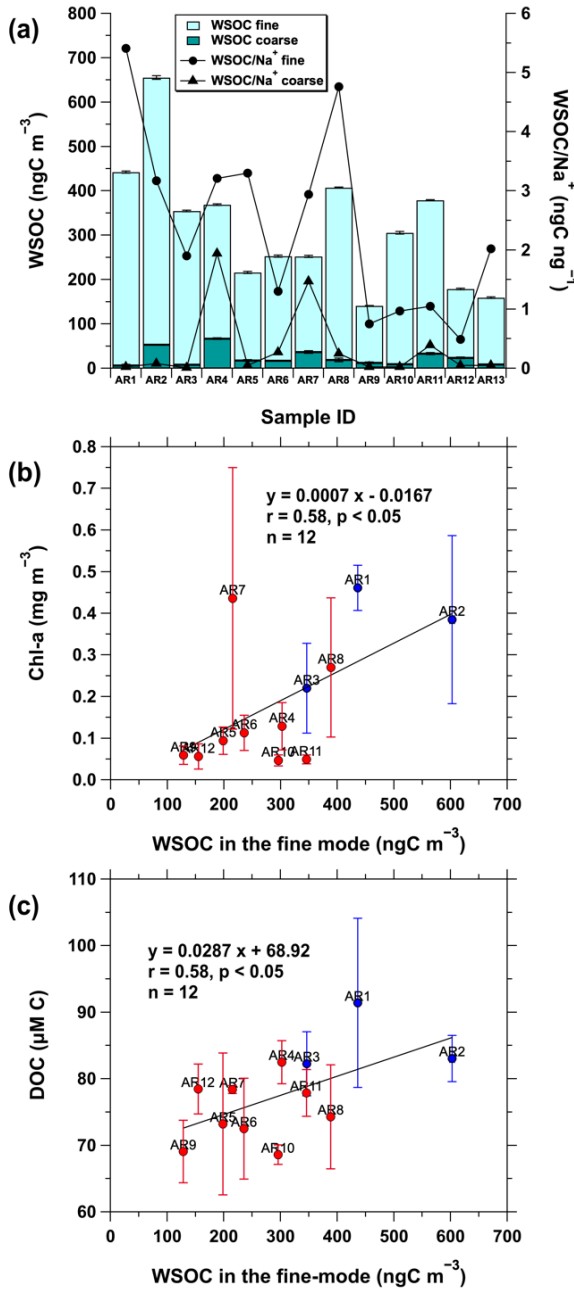

**Figure 4: (a)** Concentration of WSOC (ngC m$^{-3}$) and WSOC/Na$^+$ ratio (ngC ng$^{-1}$) in both fine- (D < 2.5 µm) and coarse-modes aerosols (2.5 µm < D < 10 µm). Relationships of WSOC concentration in the fine-mode aerosols (ngC m$^{-3}$) with **(b)** in situ surface chlorophyll-a (Chl-a) concentration (mg m$^{-3}$) and **(c)** surface dissolved organic carbon (DOC) concentration (uM C). Chl-a and DOC concentrations are presented in their mean values and standard deviations for each aerosol sampling time. The blue and red solid circles in (b) and (c) indicate the samples collected from the coastal and sea ice-covered areas, respectively. Chl-a and DOC concentrations were not measured during the collection of the AR13 aerosol sample.

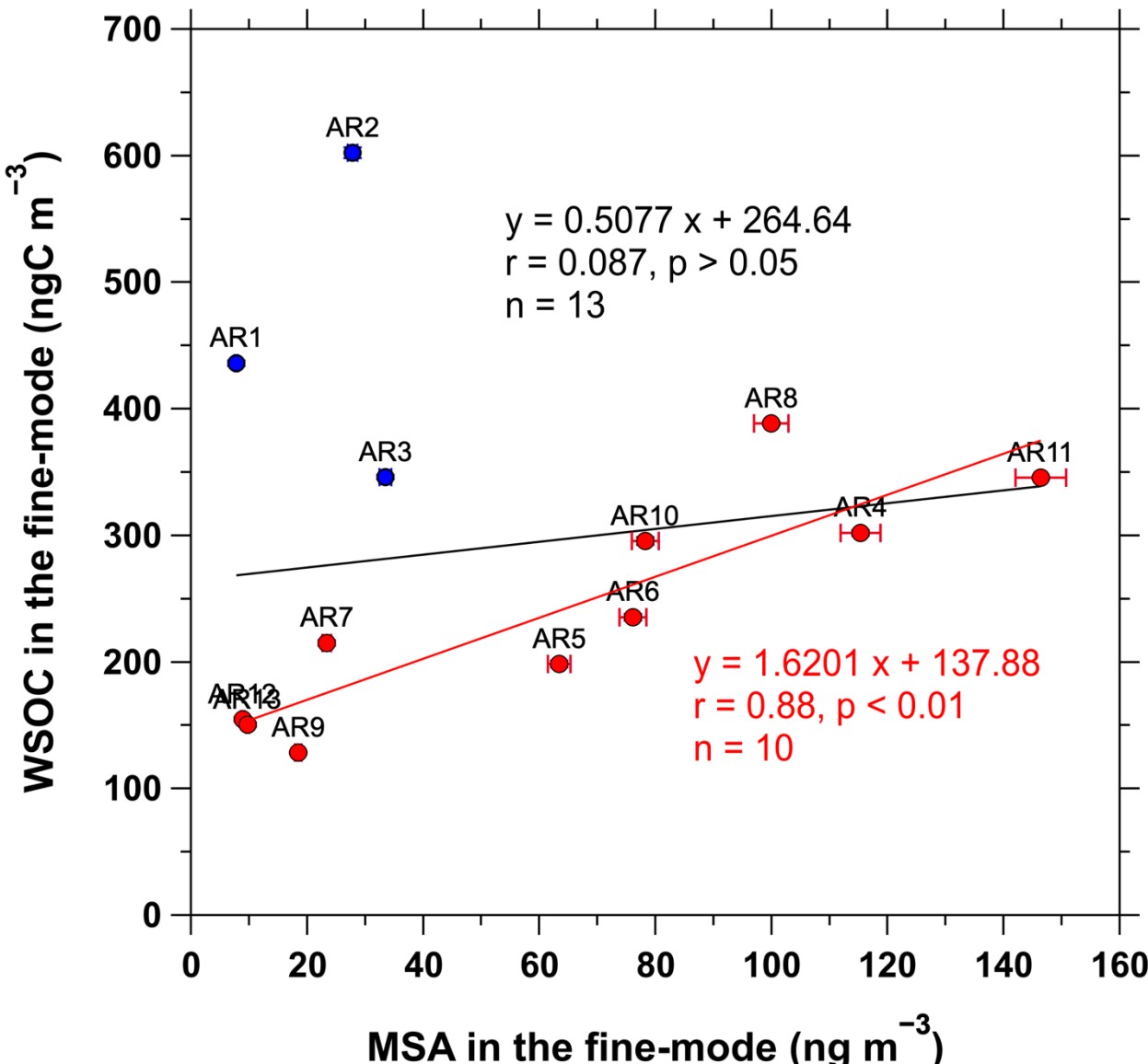

**Figure 5: Relationship between WSOC (ngC m⁻³) and MSA (ng m⁻³) in the fine-mode aerosols (D < 2.5 μm) collected during the**
**cruise. The blue and red solid circles indicate the samples collected from the coastal and sea ice-covered areas, respectively. The**
**black and red lines indicate the correlations between WSOC and MSA for the all samples collected during the cruise and for the**
**aerosol samples collected in the sea ice-covered areas of the western Arctic Ocean, respectively.**

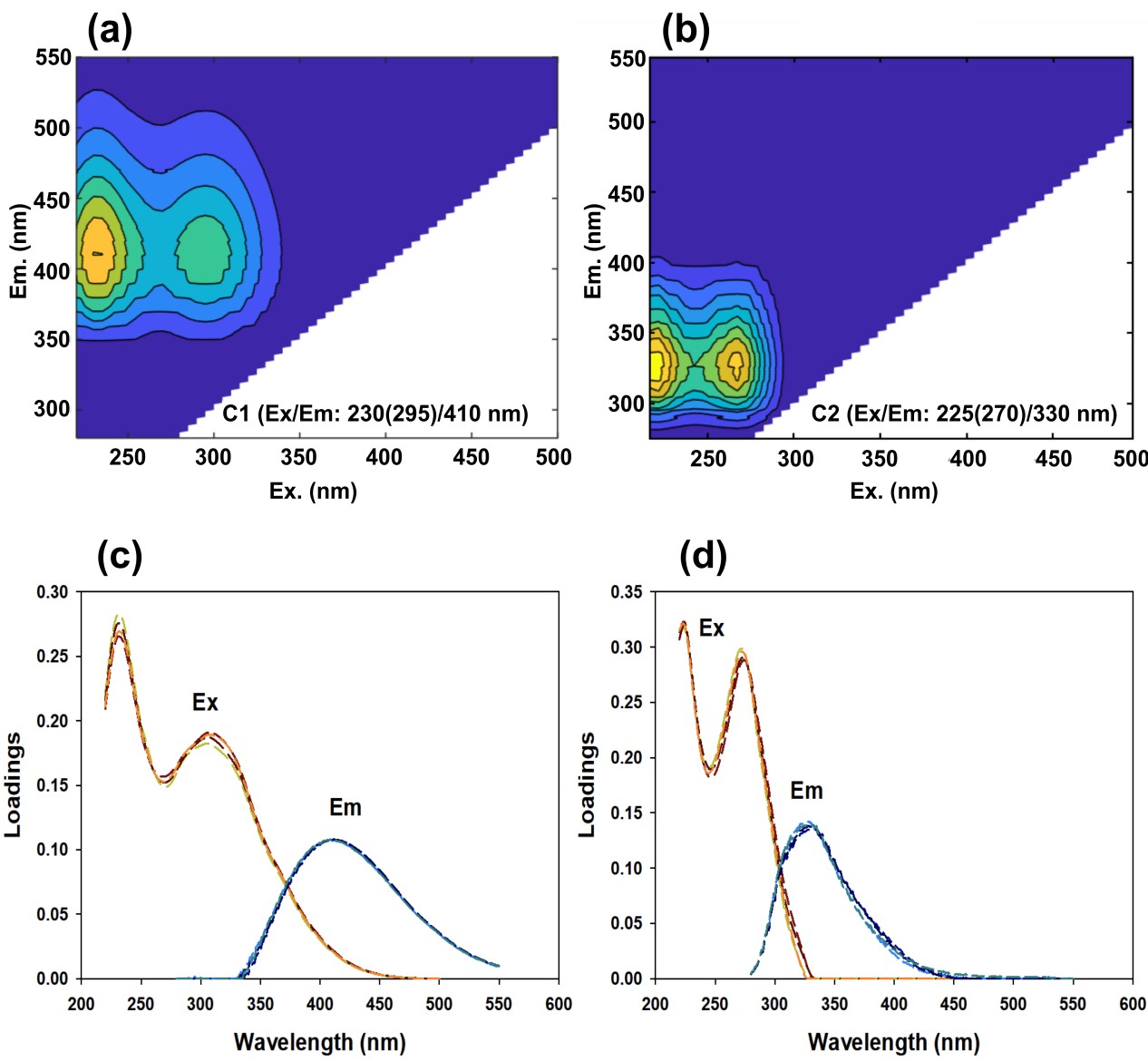

1050

Figure 6: (a and b) Fluorescence EEM contour plots of the two fluorescent components C1 and C2 identified using EEM−PARAFAC in the fine-mode aerosols collected over the western Arctic Ocean during the summer of 2016. (c and d) The loading plots of C1 and C2 showing the split-half graphs.

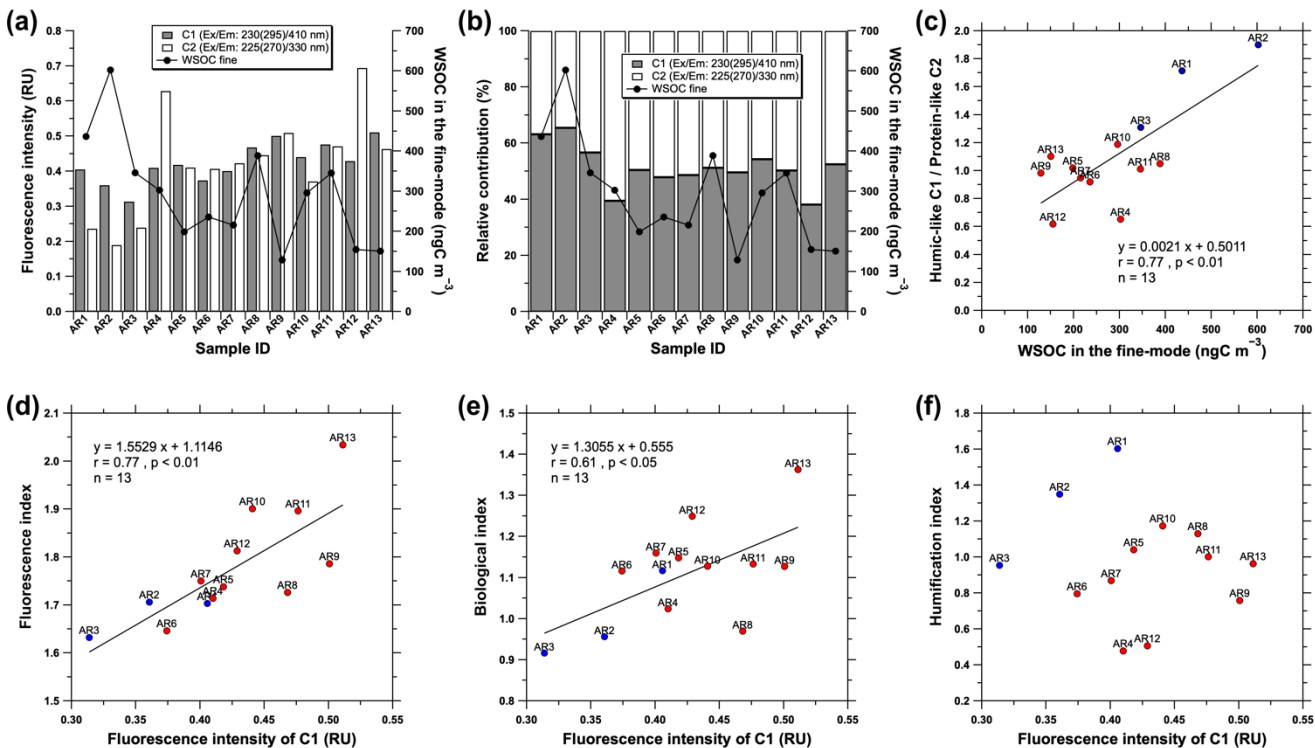

Figure 7: (a) Variations in the fluorescence intensities and (b) relative contributions of C1 and C2 in the fine-mode aerosols collected over the western Arctic Ocean during the summer of 2016. The black lines with circles in (a) and (b) indicate the WSOC concentration in the fine-mode aerosols. (c) Relationship between the fluorescence intensity ratio of the humic-like C1/protein-like C2 and the WSOC concentration in the fine-mode aerosol particles. (d) Fluorescence index (FI), (e) biological index (BIX), and (f) humification index (HIX) as a function of the fluorescence intensity of the humic-like C1 in the fine-mode aerosols. The blue and red solid circles in (c)−(f) indicate the samples collected from the coastal and sea ice-covered areas, respectively.

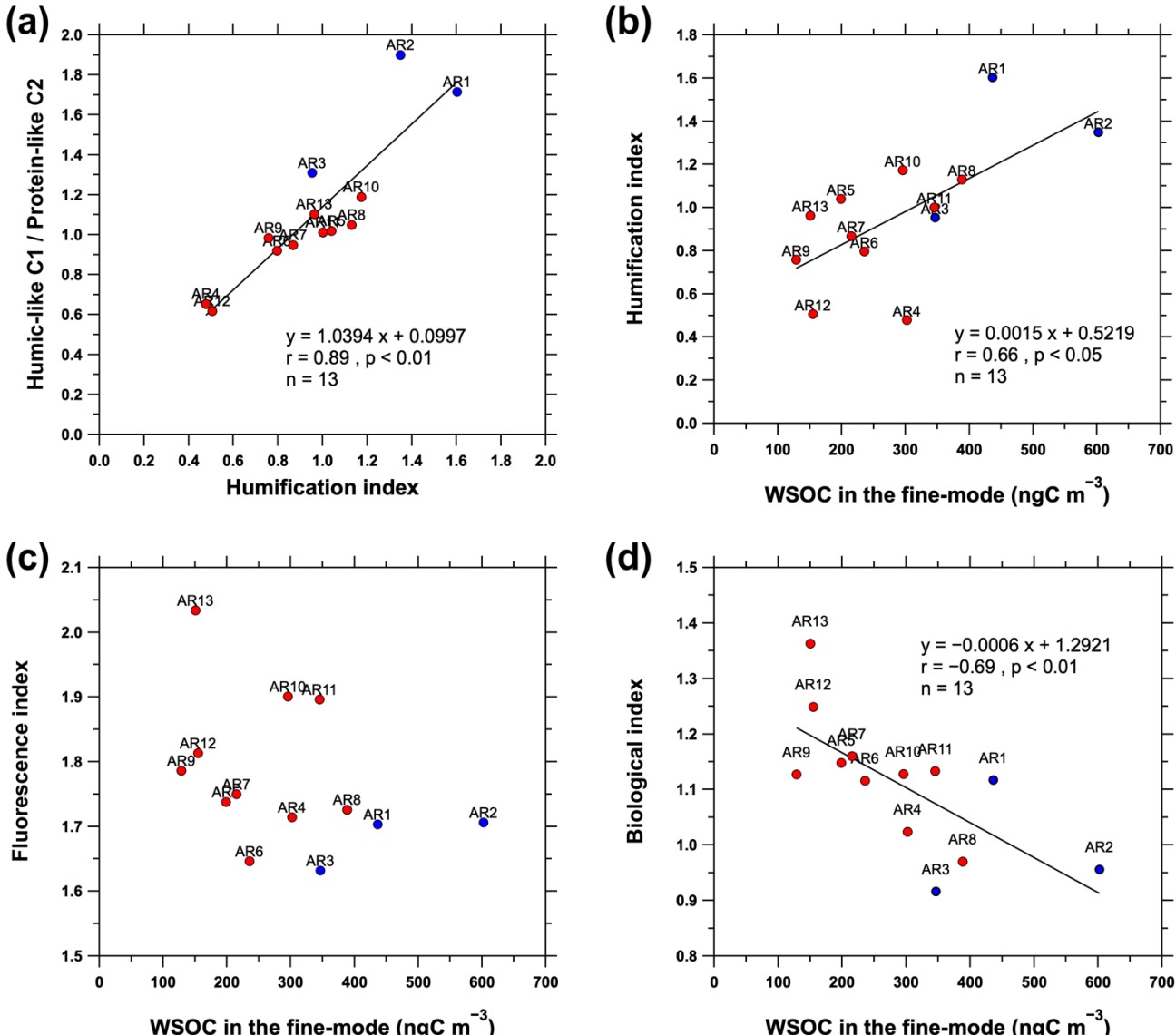

**Figure 8: (a)** Relationship between the fluorescence intensity ratio of the humic-like C1/protein-like C2 with humification index (HIX). Relationships between WSOC concentration in the fine-mode aerosols and **(b)** HIX, **(c)** fluorescence index (FI), and **(d)** biological index (BIX). The blue and red solid circles indicate the samples collected from the coastal and sea ice-covered areas, respectively.