# Peer review of "Summertime fluorescence characteristics of atmospheric watersoluble organic carbon in the marine boundary layer of the western Arctic Ocean"

_Atmospheric Chemistry and Physics, 2022_

## Referee Comment (RC1)

**General comments**

In their study, the authors focused on the fluorescence characteristics of water-soluble organic carbon (WSOC) in aerosol particles collected over the western Arctic Ocean during summer in 2016. Besides fluorescence excitation-emission matrix (EEM) coupled with parallel factor analysis (PARAFAC) of WSOC, they measured ionic species and WSOC, and marine biological parameters in surface seawaters. Two fluorescent components, humic-like C1 and protein-like C2, were identified in WSOC of fine-mode aerosols by the PARAFAC modeling. The two components varied regionally, with low fluorescent intensities in the coastal areas and high at the sea ice-covered areas.

The topic is important and actual. Although emphasize of this work is on the fluorescence characteristics of WSOC in summertime organic aerosols, a bigger part of section "Results and discussion" includes results on chemical composition of WSOC (ionic species, e.g., $Na^+$, $NO_3^-$, MSA, $nss-SO_4^{2-}$). However, I am afraid that a reader will get lost among all the results if there is not a good connection between them. Besides, nothing about these components, their importance can be found in the conclusions.
The content of this manuscript is descriptive presenting the results with some discussion, but with no discussion in the last section, where atmospheric implications should be involved; i.e. in a way how your results contribute to the understanding of the state and behavior of the atmosphere. In addition, the introduction is rather modest.

Conditionally, the manuscript could be of adequate atmospheric interest to merit publication in *Atmospheric Chemistry and Physics* (maybe as a Measurement Report), but addressing the following comments and/or questions.

**Specific comments**

Introduction:
- Line 49: The sentence "thus, WSOC is one of the most important…" can be deleted.
- Line 50: Please, correct the sentence and add some newer references. Why "residual aromatic nuclei"?
- Lines 52-54: Statement: "these discrepancies regarding the chemical composition of atmospheric WSOC« is not good. What do you mean by discrepancies? At most, you can say that there are many open questions. Besides, it is well known now that HULIS is one of the most important classes of water-soluble organics in atmospheric aerosols, fog and cloud waters. Please, correct and add some more (newer) references on HULIS in WSOC (e.g., Zheng et al., Environ. Pollut. 2013; Salma et al., J. Aerosol Sci. 2013; Frka et al., Atmos. Environ. 2018).
- Lines 59/60: Not accurate statement. Using EEM-PARAFAC approach one can get information on chemical structures and photochemical properties of compounds in

WSOC (according to their fluorescence characteristics), and based on these information you can say something on their sources and/or formation processes.

- Line 72: "…seasonal marine primary production" of what?
- Lines 81-85: The objectives of the research has to be clearly defined. Please, rewrite this paragraph.

Methods:

- It would be helpful for later discussion if the sampling sites denoted in Fig.1 are at least shortly described (e.g. AR1-AR3: coastal area)
- Except for Milli-Q water and HCl, there is no information on other chemicals/standards used.
- Lines 97-99: Combine/shorten.
- Line 197: Unit for the resistivity is $M\Omega \cdot cm$ (not $M\Omega \cdot cm^{-1}$).
- Line 110: Which separation columns did you use?
- Since sample preparation is the same for both, ionic species and WSOC, Chapters 2.1.2. and 2.1.3. can be combined.
- Line 119: Delete "In the analytical system"
- Line 120/121: Decide what to use, WSOC or TDOC. Anyway, in both cases one has in mind all water-soluble organic carbon. The sentence "Thus, in this study…" can be deleted.
- Chapter 2.1.4: Since the focus of this study are fluorescence characteristics of WSOC, this part is too superficial and has to be updated with some more information on fluorescence measurements.
- Line 145: Pre-combusted glass ampules? Were they really combusted before the use? Usually, filters for sampling need to be pre-baked to remove organic contaminants.
- Chapter 2.2.: No need to separate into 2.2.1. and 2.2.2.

Results and discussion

- Line 169: Error. Coarse aerosol above 0.5 μm?
- 3.2. Better as: "WSOC in atmospheric aerosols"
- Line 232: Correct the first line as: "The total WSOC concentration of atmospheric aerosols (fine and coarse)" (or $PM_{10}$);
- "Bulk aerosol" is not a good choice.
- Line 245, 255, 270, etc: No need to use both, OC or OM. You have to specified, what you are talking about (organic carbon - C in organic matter or organic matter).
- Lines 275-280: Be careful in comparison. Is your ratio WSOC/$Na^+$ comparable to the ratio OC/$Na^+$ (line 278/279)? Is the second also only water-soluble OC or "all" organic carbon. If so then it is logical that the ratio OC/$Na^+$ is lower than WSOC/$Na^+$.
- Lines from 335 on: Some newer references on HULIS from biomass burning and on secondary formation e.g., from phenolic precursors in multiphase reactions should be

added. Check for example: Frka et al., Atmos. Environ. 2018; Vidović et al., Environ. Sci. Technol. 2018.

Conclusions should involve atmospheric implications.
- Line 433: "One such characteristic…"? Which one?

Figure 2: Add missing information in the caption (i.e. short description of the sampling sites, e.g.: from R1 to R3: Coastal area, etc…).
Figure 4 can be moved to the Supplemental material.

---

## Referee Comment (RC2)

In this work, the fluorescent properties of water-soluble organic carbon (WSOC) present in particulate matter collected in the Western Arctic Ocean area were studied. Potential WSOC emission sources were also determined by combining parallel factor analysis (PARAFAC) and fluorescence index. On the other hand, part of the chemical composition of the collected aerosols was determined by ion chromatography.

The topic addressed in the work is relevant and current, in addition there are not many studies that use these methods to determine the sources of particulate matter, which is why it is of the utmost importance to increase knowledge of the sources that can contribute to the emission of particulate matter. WSOC in the MP.

In general terms, the work is well presented, however, the discussion of the results could be improved on some points.

Here are some recommendations in order to improve the writing:

In the Introduction section, the authors should delve into the relevance of using the EEM-PARAFAC tools in the study of particulate matter, since there are few studies on this matrix, most of which focus on the study of organic matter. dissolved in water (DOM)

In line 127 the authors could specify the type of detector of the equipment used, as well as the slit used in the measurement of both excitation and emission.

In line 133 the authors should go deeper into the objective and advantages of using fluorescence index.

Line 167. The authors should delve into how sea fog events affect the other parameters analyzed, such as TOC, among others.

Line 197. Correct resistivity units.

In section 3.2. the authors should consider changing the word "bulk" (line 232 and 236) for another that better describes the conjunction of the two fractions of the MP studied.

In section 3.4. Authors could privately benchmark their model data and compare it to a previously published dataset found in the open-access OpenFluor database located at http://openfluor.org

In section 3.4. The authors could include in the supplementary material the Split-half graphs resulting from the validation of the components of the PARAFAC models.

In the conclusions section. The authors could emphasize how their study contributes to the assignment of emission sources of the WSOC present in the PM using the EEM-PARAFAC tool.

---

## Author Response (AR1)

**Responses to Referees' comments**

21/02/2023

Journal: *Atmospheric Chemistry and Physics*

Title: Summertime fluorescence characteristics of atmospheric water-soluble organic carbon in the marine boundary layer of the western Arctic Ocean

Authors: Jinyoung Jung, Yuzo Miyazaki, Jin Hur, Yun Kyung Lee, Mi Hae Jeon, Youngju Lee, Kyoung-Ho Cho, Hyun Young Chung, Kitae Kim, Jung-Ok Choi, Catherine Lalande, Joo-Hong Kim, Taejin Choi, Young Jun Yoon, Eun Jin Yang, and Sung-Ho Kang

Manuscript number: acp-2022-665

Prof. Alex Huffman
Editor
Atmospheric Chemistry and Physics

Dear Professor Huffman,

Thank you for giving us the opportunity to submit a revised draft of the manuscript for publication in *Atmospheric Chemistry and Physics*. We appreciate the time and effort that you and the referees have dedicated to providing feedback on our manuscript. We are grateful to the referees for their insightful comments on and valuable improvements to our paper. We have been able to incorporate changes to reflect most of the suggestions provided by the referees. Additionally, the comparison of our results with the OpenFluor database and validation of the components of the PARAFAC models were conducted by Dr. Yun Kyung Lee, who was not an author in the previous manuscript. So, please accept to include her as a co-author in the revised manuscript.

We are looking forward to hearing from you in due time regarding our submission and responding to any further questions and comments you may have.

Thank you for your consideration.

Sincerely,

Jinyoung Jung

Here is a point-by-point response to the reviewers' comments and concerns.
Note: Referees' comments are highlighted in **black**, while our responses to referees are in **blue**. All modifications in the manuscript have been highlighted in yellow.

**Anonymous Referee #1:**

**General comments**

In their study, the authors focused on the fluorescence characteristics of water-soluble organic carbon (WSOC) in aerosol particles collected over the western Arctic Ocean during summer in 2016. Besides fluorescence excitation-emission matrix (EEM) coupled with parallel factor analysis (PARAFAC) of WSOC, they measured ionic species and WSOC, and marine biological parameters in surface seawaters. Two fluorescent components, humic-like C1 and protein-like C2, were identified in WSOC of fine-mode aerosols by the PARAFAC modeling. The two components varied regionally, with low fluorescent intensities in the coastal areas and high at the sea ice-covered areas.

The topic is important and actual. Although emphasize of this work is on the fluorescence characteristics of WSOC in summertime organic aerosols, a bigger part of section "Results and discussion" includes results on chemical composition of WSOC (ionic species, e.g., $Na^+$, $NO_3^-$, MSA, nss-$SO_4^{2-}$). However, I am afraid that a reader will get lost among all the results if there is not a good connection between them. Besides, nothing about these components, their importance can be found in the conclusions.

The content of this manuscript is descriptive presenting the results with some discussion, but with no discussion in the last section, where atmospheric implications should be involved; i.e. in a way how your results contribute to the understanding of the state and behavior of the atmosphere. In addition, the introduction is rather modest.

Conditionally, the manuscript could be of adequate atmospheric interest to merit publication in Atmospheric Chemistry and Physics (maybe as a Measurement Report), but addressing the following comments and/or questions.

(Response) We greatly appreciate Referee #1's careful review and thoughtful comments. According to Referee #1's insightful comments, we have carefully revised the manuscript and provided point-by-point responses. We hope that the correction will meet your high standards.

**Specific comments**

Introduction:

Line 49: The sentence "thus, WSOC is one of the most important…" can be deleted.

(Response) We greatly appreciate Referee #1's insightful comment. According to Reviewer #1's comment, we have removed "thus, WSOC is one of the most important carbon components" from the sentence (page 2, line 48).

Line 50: Please, correct the sentence and add some newer references. Why "residual aromatic nuclei"?

(Response) We greatly appreciate Referee #1's insightful comment. We referred to Decesari et al. (2001) to describe the composition of atmospheric water-soluble organic carbon (WSOC). Decesari et al. (2001) investigated the main structural features of aerosol water-soluble organic carbon (WSOC) using ion exchange chromatography (IEC) and proton nuclear magnetic resonance (HNMR). Based on their results, Decesari et al. (2001) reported that WSOC is composed of highly oxidized species with residual aromatic nuclei and aliphatic chains. Thus, we described that "WSOC comprises highly oxidized species with residual aromatic nuclei and aliphatic chains (Decesari et al., 2001)". However, now we have realized that the sentence is needed to be revised, as Referee #1 pointed out. Thus, we have revised the sentence by adding new references as follows:

"Detailed chemical analysis of atmospheric WSOC revealed that acidic compounds, including monoacids, diacids, and polyacidic compounds, are a major fraction of WSOC. (e.g., Decesari et al., 2001; Cavalli et al., 2004; Sullivan et al., 2004; Psichoudaki and Pandis, 2013; Xie et al., 2016). In particular, polyacidic compounds, which are composed of aromatic compounds that have aliphatic chains with oxygenated functional groups (e.g., carboxyl, hydroxyl, and carbonyl groups), are often referred to as humic-like substances (HULIS) (Decesari et al., 2001; Kiss et al., 2002; Graber and Rudich, 2006; Salma et al., 2013)." (page 2, line 49−54).

Lines 52-54: Statement: "these discrepancies regarding the chemical composition of atmospheric WSOC« is not good. What do you mean by discrepancies? At most, you can say that there are many open questions. Besides, it is well known now that HULIS is one of the most important classes of water-soluble organics in atmospheric aerosols, fog and cloud waters. Please, correct and add some more (newer) references on HULIS in WSOC (e.g., Zheng et al., Environ. Pollut. 2013; Salma et al., J. Aerosol Sci. 2013; Frka et al., Atmos. Environ. 2018).

(Response) We greatly appreciate Referee #1's insightful comment. As described in the unrevised manuscript, the chemical nature of 10−20% of WSOC has been resolved at a molecular level, although WSOC is a major constituent of organic carbon (OC) in aerosols. The major composition of WSOC remains chemically unresolved. Thus, we used "discrepancies" to describe the lack of comprehensive information regarding the chemical composition of atmospheric WSOC. However, now we have realized that the descriptions should be revised, as Referee #1 pointed out. Thus, we have revised the descriptions by adding new references as follows:

"HULIS are present ubiquitously in aerosol particles from various environments (e.g., urban, rural, forest, and marine) (e.g., Cavalli et al., 2004; Graber and Rudich, 2006; Hoffer et al., 2006; Fu et al., 2015; Chen et al., 2016; Fan et al., 2016; Frka et al., 2018), fog (Krivácsy et al., 2000), rain (Kieber et al., 2006; Yang et al., 2019), and snow (Voisin et al., 2012). In addition, HULIS constitute a significant fraction (25−75%) of aerosol WSOC (Zheng et al., 2013). However, despite a large number of studies on the investigation of individual or classes of compounds, as mentioned above, complete molecular-level chemical characterization of the WSOC remains currently unavailable (Zheng et al., 2013; Fu et al., 2015). Thus, the lack of comprehensive information

regarding the chemical composition of atmospheric WSOC hinders a deeper understanding of the role of WSOC in aerosol characteristics." (page 2, line 54−61).

Lines 59/60: Not accurate statement. Using EEM-PARAFAC approach one can get information on chemical structures and photochemical properties of compounds in WSOC (according to their fluorescence characteristics), and based on these information you can say something on their sources and/or formation processes.
(Response) We greatly appreciate Referee #1's insightful comment. We agree with this comment. According to Referee #1's insightful comment, we have revised the sentence as follows:

"Moreover, recent studies have demonstrated that EEM–PARAFAC is a useful tool to reveal the optical properties and chemical structures of atmospheric WSOC, which provide information regarding its origins, chemical reactions, and formation processes" (page 3, line 69−71).

Line 72: "…seasonal marine primary production" of what?
(Response) We greatly appreciate Referee #1's insightful comment. According to Referee #1's insightful comment, we have revised "…seasonal marine primary production" to "…seasonal primary production of marine phytoplankton" (page 3, line 83−84).

Lines 81-85: The objectives of the research has to be clearly defined. Please, rewrite this paragraph.
(Response) We greatly appreciate Referee #1's insightful comment. According to Referee #1's insightful comment, we have revised the paragraph as follows:

"In this study, simultaneous measurements of aerosol chemical composition and fluorescence properties of WSOC, together with a hydrographic survey, were carried out in the western Arctic Ocean during the summer of 2016 to improve our understanding of the characteristics of atmospheric WSOC. Accordingly, this study aimed to (1) investigate the distributions of ionic species, which could potentially provide useful information for characterizing sources (e.g., sea spray, biogenic, and anthropogenic) and formation mechanisms (i.e., primary and secondary processes) of WSOC, (2) examine the characteristics of atmospheric WSOC using ionic species and hydrographic data, and (3) characterize the quality and possible formation pathways of atmospheric WSOC with fluorescence EEM–PARAFAC." (page 3, line 92−98).

Methods:
It would be helpful for later discussion if the sampling sites denoted in Fig.1 are at least shortly described (e.g. AR1-AR3: coastal area)
(Response) We greatly appreciate Referee #1's insightful comment. According to Referee #1's insightful comment, we have added the following description to the Method.

"For the purpose of regional analysis, the study area was geographically divided into two regions: the coastal area (aerosol samples AR1−AR3) and the sea ice-covered area (aerosol samples AR4−AR13)." (page 4, line 104−106).

Except for Milli-Q water and HCl, there is no information on other chemicals/standards used.

(Response) We greatly appreciate Referee #1's insightful comment. According to Referee #1's insightful comment, we have added the descriptions of other chemicals and standards to Methods (page 4, line 125− page 5, line 130; page 5, line 139−140).

Lines 97-99: Combine/shorten.

(Response) We greatly appreciate Referee #1's insightful comment. According to Referee #1's insightful comment, we have revised the sentences as follows:

"A wind-sector controller allowed aerosol samples to be collected only when the relative wind directions were within ±100°, relative to the ship's bow and when the relative wind speeds were > 1 m s$^{-1}$ (Jung et al., 2020)." (page 4, line 113−115).

Line 197: Unit for the resistivity is MΩ·cm (not MΩ·cm-1).

(Response) We greatly appreciate Referee #1's insightful comment. According to Reviewer #1's comment, we have revised "MΩ cm$^{-1}$" to "MΩ·cm" (page 4, line 122).

Line 110: Which separation columns did you use?

(Response) We greatly appreciate Referee #1's insightful comment. According to Referee #1's insightful comment, we have added the following descriptions of ion separation columns to Section 2.1.2.

"Anions were analyzed using an AS19 anion exchange column (Thermo Scientific Dionex). An eluent generator equipped with an EGC-KOH cartridge was used to produce potassium hydroxide eluent. Cations were separated and quantified using a CS12A cation exchange column (Thermo Scientific Dionex). A solution of MSA (99.0% purity, Sigma-Aldrich) was used as the eluent for the cations. Calibrations were conducted using multilevel standard solutions diluted with stock solutions from Thermo Scientific Dionex (anion P/N 057590 and cation P/N 046070)." (page 4, line 125−page 5, line 130).

Since sample preparation is the same for both, ionic species and WSOC, Chapters 2.1.2. and 2.1.3. can be combined.

(Response) We greatly appreciate Referee #1's insightful comment. According to Referee #1's insightful comment, we have combined Section 2.1.2 with Section 2.1.3. Thus, we have revised the title of Section 2.1.2 to "2.1.2 Chemical analyses of ionic species and water-soluble organic carbon" (page 4, line 119) and have changed Section number "2.1.4" to "2.1.3" (page 5, line 144).

Line 119: Delete "In the analytical system"

(Response) We greatly appreciate Referee #1's insightful comment. According to Referee #1's insightful comment, we have removed "In the analytical system" from the sentence (page 5, line 138).

Line 120/121: Decide what to use, WSOC or TDOC. Anyway, in both cases one has in mind all water-soluble organic carbon. The sentence "Thus, in this study…" can be deleted.

(Response) We greatly appreciate Referee #1's insightful comment. We have decided to use WSOC, according to Referee #1's insightful comment. Thus, we have revised the sentences as follows:

"The resultant filtrates were analyzed for WSOC using a total organic carbon (TOC) analyzer (model TOC-L, Shimadzu Inc., Japan). Inorganic carbon was removed by acidifying the samples to pH 2 using 2 M HCl and subsequent sparging for 8 min before conducting the WSOC analysis." (page 5, line 136−139).

Additionally, as suggested, we have removed "Thus, in this study, the TDOC in the filtrates was defined as WSOC (Jung et al., 2020)." from the paragraph (page 5, line 139).

Chapter 2.1.4: Since the focus of this study are fluorescence characteristics of WSOC, this part is too superficial and has to be updated with some more information on fluorescence measurements.
(Response) We greatly appreciate Referee #1's insightful comment. According to Referee #1's insightful comment, we have revised the descriptions of fluorescence measurement as follows:

"Three-dimensional fluorescence EEMs were measured using a luminescence spectrometer (Hitachi F-7000, Hitachi Inc., Japan) equipped with a light source of 150 W xenon lamp. The wavelength range of the scanning was set at 250−500 nm for excitation (Ex) with a 5 nm step and 280−550 nm for emission (Em) with a 1 nm step (Jung et al., 2020). The slits for both Ex and Em were fixed at 10 nm. The EEMs of each sample were calibrated by subtracting the EEM of Milli-Q water and were normalized to Raman unit (RU) by integrating the Raman bands from 380 to 420 nm at a 350 nm excitation (Lawaetz and Stedmon, 2009; Stedmon et al., 2003; Chen et al., 2018). Before calibration, inner filter effects were corrected using absorbance spectra of the same sample (McKnight et al., 2001). Absorbance spectra were measured on an ultraviolet-visible spectrophotometer (Shimadzu 1800, Shimadzu Inc., Japan) using a 1 cm quartz cuvette. The sample EEMs were compiled and characterized by PARAFAC using MATLAB 7.0.4 with the DOMFluor toolbox (Stedmon and Bro, 2008). The number of fluorescent components was determined based on split-half validation and the percentage of the explained variance (99.3%). The loadings in the Ex and Em for each component were matched to the OpenFluor database with more than 93% similarity (Table 1) (Murphy et al., 2014). The humification index (HIX, ratio of emission intensity 435−480 nm/300−345 nm at 255 nm excitation; Zsolnay et al., 1999), biological index (BIX, ratio of emission intensity 380 nm/430 nm at 310 nm excitation; Huguet et al., 2009), and fluorescence index (FI, ratio of emission intensity 450 nm/500 nm at 370 nm excitation; McKnight et al., 2001) were calculated from the EEMs. These fluorescence indices have been widely applied in studies of aquatic and terrestrial environments because they provide insights into the sources and chemical properties of chromophores (Zsolnay et al., 1999; McKnight et al., 2001; Huguet et al., 2009). Moreover, previous studies (e.g., Lee et al., 2013; Fu et al., 2015; Chen et al., 2016; Tang et al., 2021; Wu et al., 2021) revealed that the HIX, BIX, and FI can provide useful information on the degree of humification, the chemical structures, and aging processes of atmospheric WSOC, as detailed in Sect. 3.4." (page 5, line 146−page 6, line 164).

Line 145: Pre-combusted glass ampules? Were they really combusted before the use? Usually, filters for sampling need to be pre-baked to remove organic contaminants.

(Response) We greatly appreciate the valuable question. During the preparation for seawater sampling in our land laboratory, we combusted glass ampoules as well as Whatman GF/F filters at 550 °C for 6 hours in a muffle furnace to eliminate any organic contaminants. We used pre-combusted glass ampoules and Whatman GF/F filters to collect seawater samples for dissolved organic carbon (DOC) during the cruise. In addition, as described in Section 2.2.1, we believe that the meaning of "pre-combusted" is identical to that of "pre-baked" mentioned by Referee #1.

Chapter 2.2.: No need to separate into 2.2.1. and 2.2.2.

(Response) We greatly appreciate Referee #1's insightful comment. According to Referee #1's insightful comment, we have combined Section 2.2.1 with Section 2.2.2. Thus, we have revised the title of Section 2.2.1 to "2.2.1 Dissolved organic carbon and in situ chlorophyll-a" (page 6, line 169).

Results and discussion
Line 169: Error. Coarse aerosol above 0.5 μm?

(Response) We greatly appreciate Referee #1's insightful comment. According to Referee #1's insightful comment, we have revised "coarse aerosols (D > 0.5 μm)" to "coarse aerosols D > 2.5 μm)" (page 7, line 196−197).

3.2. Better as: "WSOC in atmospheric aerosols"

(Response) We greatly appreciate Referee #1's insightful comment. According to Referee #1's insightful comment, we have revised the title of Section 3.2 "Atmospheric WSOC concentration" to "3.2 WSOC in atmospheric aerosols" (page 9, line 258).

Line 232: Correct the first line as: "The total WSOC concentration of atmospheric aerosols (fine and coarse)" (or PM10);

(Response) We greatly appreciate Referee #1's insightful comment. According to Referee #1's insightful comment, we have revised "The atmospheric WSOC concentration in bulk (fine + coarse) aerosols…." to "The total WSOC concentration of atmospheric aerosols (fine and coarse)…." (page 9, line 259).

"Bulk aerosol" is not a good choice.

(Response) We greatly appreciate Referee #1's insightful comment. According to Referee #1's insightful comment, we have revised "Further, the mean WSOC concentration in bulk aerosols in the western Arctic Ocean…." to "Further, the mean total WSOC concentration observed in the western Arctic Ocean…." (page 9, line 269).

Line 245, 255, 270, etc: No need to use both, OC or OM. You have to specified, what you are talking about (organic carbon - C in organic matter or organic matter).

(Response) We greatly appreciate Referee #1's insightful comment. We agree with this comment. In the unrevised manuscript, we used "OM" as the abbreviation for "organic mass" not for "organic matter". However, now we have realized that it could cause a misunderstanding. Thus, we have taken Referee #1's insightful comment by revising "OC

(or OM)" to "OC" (page 9, line 278; page 10, line 288, 303, 305; page 1l, line 321) and "OM" to "organic mass" throughout the manuscript (page 9, line 276; page 10, line 319; page 10, line 320−page 11, line 321).

Lines 275-280: Be careful in comparison. Is your ratio WSOC/Na+ comparable to the ratio OC/Na+ (line 278/279)? Is the second also only water-soluble OC or "all" organic carbon. If so then it is logical that the ratio OC/Na+ is lower than WSOC/Na+.
(Response) We greatly appreciate the valuable question. The OC reported by Russell et al. (2010) and Frossard et al. (2014) represents the total OC derived from FTIR measurements of organic mass. Thus, we believe that it is reasonable to describe as follows:

"However, the WSOC/Na$^+$ ratios of the fine-mode aerosols observed in this study were higher than those measured previously in submicron primary marine aerosol particles having OC/Na$^+$ ranging from 0.1 to 2 (Russell et al., 2010; Frossard et al., 2014)." (page 10, line 310−312).

Lines from 335 on: Some newer references on HULIS from biomass burning and on secondary formation e.g., from phenolic precursors in multiphase reactions should be added. Check for example: Frka et al., Atmos. Environ. 2018; Vidović et al., Environ. Sci. Technol. 2018.
(Response) We greatly appreciate Referee #1's insightful comment. According to Referee #1's insightful comment, we have added the following references to the descriptions of the sources of humic-like substances in aerosols (page 12, line 371−372, 375).

Barsotti, F., Ghigo, G., and Vione, D.: Computational assessment of the fluorescence emission of phenol oligomers: a possible insight into the fluorescence properties of humic-like substances (HULIS). J. Photochem. Photobiol., A 315, 87–93. https://doi.org/10.1016/j.jphotochem.2015.09.012, 2016.

Frka, S., Grgić, I., Turšič, J., Gini, M. I., and Eleftheriadis, K.: Seasonal variability of carbon in humic-like matter of ambient size-segregated water soluble organic aerosols from urban background environment, Atmos. Environ., 173, 239–247, https://doi.org/10.1016/j.atmosenv.2017.11.013, 2018.

Pani, S. K., Lee, C. -T., Griffth, S. M., and Lin, N. -H.: Humic-like substances (HULIS) in springtime aerosols at a high-altitude background station in the western North Pacific: Source attribution, abundance, and light-absorption, Sci. Total Environ., 809, 151180, https://doi.org/10.1016/j.scitotenv.2021.151180, 2022.

Qin, J., Zhang, L., Zhou, X., Duan, J., Mu, S., Xiao, K., Hu, J., and Tan, J.: Fluorescence fingerprinting properties for exploring water-soluble organic compounds in PM2.5 in an industrial city of northwest China, Atmos. Environ., 184, 203–211, https://doi.org/10.1016/j.atmosenv.2018.04.049, 2018.

Vidović, K., Jurković, D. L., Šala, M., Kroflič, A., and Grgić, I.: Nighttime aqueous-phase formation of nitrocatechols in the atmospheric condensed phase, Environ. Sci.

Technol., 52, 9722–9730, https://doi.org/10.1021/acs.est.8b01161, 2018.

Conclusions should involve atmospheric implications.
(Response) We greatly appreciate Referee #1's insightful comment. According to Referee #1's insightful comment, we have improved the manuscript as follows:

"Recent hydrographic changes associated with changes in seasonal primary production of marine phytoplankton in response to recent sea ice loss and increased wind mixing have been reported in the Arctic Ocean (Ardyna and Arrigo, 2020; Lewis et al., 2020), which can influence on the productions of primary organic aerosols and secondary organic aerosol precursors that impact aerosol composition and size (Lannuzel et al., 2020). These changes will induce alterations in the quantity and quality of atmospheric WSOC in the Arctic Ocean. In addition, a shift in aerosol chemistry toward more oxygenated and smaller molecules has been reported as the available solar radiation increases after polar sunrise (Fu et al., 2009). Further, the dominance of more oxygenated organic species in Arctic spring results in a more hygroscopic organic component compared to that observed during cleaner times (Willis et al., 2018 and references therein). Consequently, the chemical properties of relatively new and less oxygenated WSOC in the sea ice-covered area are expected to change due to photochemical oxidation processes, altering the hygroscopic properties of WSOC in the Arctic Ocean during summer. Further studies are therefore required to more clearly understand the characteristics of atmospheric OC in the rapidly changing Arctic Ocean." (page 16, line 497−507).

Line 433: "One such characteristic…"? Which one?
(Response) We greatly appreciate Referee #1's insightful comment. Now we have realized that the sentence is needed to be revised, as Referee #1 pointed out. According to Referee #1's insightful comment, we have improved the manuscript as follows:

"The WSOC concentration observed in the western Arctic Ocean (range = 141–656 ngC m$^{-3}$, mean = 316 ± 141 ngC m$^{-3}$) was substantially higher than in the Amundsen Sea in West Antarctica (range = 70–180 ngC m$^{-3}$, mean = 97 ± 38 ngC m$^{-3}$; Jung et al., 2020), where massive phytoplankton blooms are present (Arrigo et al., 2012). However, a clear distinction in $NO_3^-$ concentration between coastal (158 ± 130 ng m$^{-3}$) and sea ice-covered areas (10 ± 4.9 ng m$^{-3}$) and the MSA/nss-$SO_4^{2-}$ ratios in the fine-mode aerosols (0.21 ± 0.16) comparable to the other summertime values measured in the central Arctic Ocean (0.22, Leck and Persson, 1996; 0.25 ± 0.02, Chang et al., 2011) suggested that the influence of continental sources was minimal, especially in the sea ice-covered areas of the western Arctic Ocean during summer, although it cannot be excluded. In addition, the positive correlations between WSOC concentration, WSOC/$Na^+$ ratio in the fine-mode aerosols, in situ surface Chl-a, and surface DOC concentrations suggested that the WSOC in aerosols was influenced by the sea-to-air transfer of the oceanic source of OC partially along with biological productivity. During the sampling period, six aerosol samples were largely or slightly affected by sea fog events, but no significant differences in mean total WSOC concentrations were observed between sea fog (328 ± 112 ngC m$^{-3}$) and non-sea fog events periods (307 ± 171 ngC m$^{-3}$). This result reflects that WSOC in aerosols was

less likely affected by the preferential scavenging processes of coarse particles by sea fog than $Na^+$ and $NO_3^-$. However, sea fog could contribute to the formation of atmospheric WSOC, making favorable conditions for secondary processes due to high relative humidity conditions, secondary formation by heterogeneous reactions and/or oligomerization through in-cloud processing. Furthermore, the relationship between WSOC and MSA in the fine-mode aerosols revealed a strong connection of WSOC with secondary processes in the sea ice-covered areas, probably due to greater sea–air gas exchange and DMS and VOC emissions because of sea ice retreat and increasing available solar radiation during the Arctic summer." (page 15, line 463−481).

Figure 2: Add missing information in the caption (i.e. short description of the sampling sites, e.g.: from R1 to R3: Coastal area, etc…).
(Response) We greatly appreciate Referee #1's insightful comment. According to Referee #1's insightful comment, we have revised the caption of Figure 2 as follows:

"Figure 1: Concentrations of (a) $Na^+$ (ng m$^{-3}$), (b) $NO_3^-$ (ng m$^{-3}$), (c) MSA (ng m$^{-3}$), and (d) nss-$SO_4^{2-}$ (ng m$^{-3}$) in aerosols collected from the coastal (AR1–AR3) and sea ice-covered areas (AR4–AR13) of the western Arctic Ocean during the summer of 2016." (page 33, line 1007−1008).

Figure 4 can be moved to the Supplemental material.
(Response) We greatly appreciate Referee #1's insightful comment. According to Referee #1's insightful comment, we have moved Figure 4 to the Supplementary material. We also have changed the Figure numbers throughout the manuscript (e.g., Figure 5 to Figure 4).

**Anonymous Referee #2:**

In this work, the fluorescent properties of water-soluble organic carbon (WSOC) present in particulate matter collected in the Western Arctic Ocean area were studied. Potential WSOC emission sources were also determined by combining parallel factor analysis (PARAFAC) and fluorescence index. On the other hand, part of the chemical composition of the collected aerosols was determined by ion chromatography. The topic addressed in the work is relevant and current, in addition there are not many studies that use these methods to determine the sources of particulate matter, which is why it is of the utmost importance to increase knowledge of the sources that can contribute to the emission of particulate matter. WSOC in the MP. In general terms, the work is well presented, however, the discussion of the results could be improved on some points.
(Response) We greatly appreciate Referee #2's careful review and thoughtful comments. According to Referee #2's insightful comments, we have carefully revised the manuscript and provided point-by-point responses. We hope that the correction will meet your high standards.

Here are some recommendations in order to improve the writing:

In the Introduction section, the authors should delve into the relevance of using the EEMPARAFAC tools in the study of particulate matter, since there are few studies on this matrix, most of which focus on the study of organic matter. dissolved in water (DOM)

(Response) We greatly appreciate Referee #2's insightful comment. According to Referee #2's insightful comment, we have improved the manuscript as follows:

"Chromophoric dissolved organic matter (CDOM) in aquatic environments is relatively well characterized by EEM–PARAFAC (e.g., Coble, 2007; Ishii and Boyer, 2012), whereas it has not been extensively used for the analysis of organic matter in atmospheric aerosols (Mladenov et al., 2011). Chromophore components associated with HULIS and protein-like substances have been determined for WSOC in atmospheric aerosols, suggesting the potential of EEM–PARAFAC in atmospheric analysis (Duarte et al., 2004). Moreover, recent studies have demonstrated that EEM–PARAFAC is a useful tool to reveal the optical properties and chemical structures of atmospheric WSOC, which provide information regarding its origins, chemical reactions, and formation processes" (page 3, line 65−71).

In line 127 the authors could specify the type of detector of the equipment used, as well as the slit used in the measurement of both excitation and emission.

(Response) We greatly appreciate Referee #2's insightful comment. The detail of the instrument has been supplemented as follows:

"Three-dimensional fluorescence EEMs were measured using a luminescence spectrometer (Hitachi F-7000, Hitachi Inc., Japan) equipped with a light source of 150 W xenon lamp. The wavelength range of the scanning was set at 250−500 nm for excitation (Ex) with a 5 nm step and 280−550 nm for emission (Em) with a 1 nm step (Jung et al., 2020). The slits for both Ex and Em were fixed at 10 nm. The EEMs of each sample were calibrated…" (page 5, line 146−149).

In line 133 the authors should go deeper into the objective and advantages of using fluorescence index.

(Response) We greatly appreciate Referee #2's insightful comment. According to Referee #2's insightful comment, we have improved the manuscript by adding the following descriptions to Sect. 2.1.3.

"These fluorescence indices have been widely applied in studies of aquatic and terrestrial environments because they provide insights into sources and chemical properties of chromophores (Zsolnay et al., 1999; McKnight et al., 2001; Huguet et al., 2009). Moreover, previous studies (e.g., Lee et al., 2013; Fu et al., 2015; Chen et al., 2016; Tang et al., 2021; Wu et al., 2021) revealed that the HIX, BIX, and FI can provide useful information on the degree of humification, the chemical structures, and aging processes of atmospheric WSOC, as detailed in Sect. 3.4." (page 5, line 160−page 6, line 164).

Line 167. The authors should delve into how sea fog events affect the other parameters analyzed, such as TOC, among others.

(Response) We greatly appreciate Referee #2's insightful comment. According to Referee #2's insightful comment, we have improved the manuscript by adding the following descriptions to Section 3.2.

"The mean total WSOC concentrations observed during sea fog events (AR1, AR3, AR4, AR6, AR8, and AR9; 328 ± 112 ngC m$^{-3}$) were comparable to those during non-sea fog events (AR2, AR5, AR7, AR10−AR13; 307 ± 171 ngC m$^{-3}$), reflecting that WSOC was less likely affected by the preferential scavenging processes of coarse particles by sea fog than Na$^+$ and NO$_3^-$. Although the influence of sea fog on WSOC concentration in aerosols was not particularly remarkable in this study, it is worth mentioning that sea fog could contribute to the formation of atmospheric WSOC, making favorable conditions for secondary processes (e.g., condensation of organic species on pre-existing aerosol particles) (Blando and Turpin, 2000; Kanakidou et al., 2005; Ervens et al., 2011)." (page 9, line 262−268).

Line 197. Correct resistivity units.
(Response) We greatly appreciate Referee #2's insightful comment. According to Reviewer #1's comment, we have revised "MΩ cm$^{-1}$" to "MΩ·cm" (page 4, line 122).

In section 3.2. the authors should consider changing the word "bulk" (line 232 and 236) for another that better describes the conjunction of the two fractions of the MP studied.
(Response) We greatly appreciate Referee #2's insightful comment. According to Referee #2's insightful comment, we have revised "The atmospheric WSOC concentration in bulk (fine + coarse) aerosols…." to "The total WSOC concentration of atmospheric aerosols (fine and coarse)…." (page 9, line 259) and "Further, the mean WSOC concentration in bulk aerosols in the western Arctic Ocean…." to "Further, the mean total WSOC concentration observed in the western Arctic Ocean….", respectively (page 9, line 269).

In section 3.4. Authors could privately benchmark their model data and compare it to a previously published dataset found in the open-access OpenFluor database located at http://openfluor.org
(Response) We greatly appreciate Referee #2's insightful comment. According to Referee #2's insightful comment, we have modified the related text and supplemented more information regarding the OpenFluor database along with Table 1 as follows:

1. "The number of fluorescent components was determined based on split-half validation and the percentage of the explained variance (99.3%). The loadings in the Ex and Em for each component were matched to the OpenFluor database with more than 93% similarity (Table 1) (Murphy et al., 2014)." (page 5, line 154−157).

2. Table 1. Excitation (Ex) and emission (Em) maxima of the two fluorescent components, their assignments, and the comparison with previous literatures (page 31).

| Components | Ex. (nm) | Em. (nm) | Assignments (Labeled by Coble) | Literature Comparison |
|---|---|---|---|---|

| | | | | |
|---|---|---|---|---|
| C1 | 230(295) | 410 | Marine Humic-like (Combination of traditionally defined peak A and M) | C4: <260(305)/404 [Chukchi Seawater] (Chen et al., 2018) |
| | | | | C1: 305/410 [Beaufort Seawater] (Gao and Guéguen, 2017) |
| | | | | C4: 295/405 [Svalbard fjord Seawater] (Brogi et al., 2019) |
| | | | | C1: 300/416 [Greenland Ice core] (D'Andrilli and McConnell, 2021) |
| C2 | 225(270) | 330 | Protein-like (Traditionally defined peak T) | C2: 275/338 [Chukchi Seawater] (Chen et al., 2018) |
| | | | | C3: 273/332 [Greenland Seawater] (Goncalves-Araujo et al., 2016) |
| | | | | C4: 275/320 [Beaufort Seawater] (Dainard et al., 2015) |
| | | | | C5: 240(280)/322 [Greenland Lake] (Osburn et al., 2017) |

The comparison is based on the similarity >93% obtained using the OpenFluor database.

3. "These fluorescent components were generally comparable to those previously observed in the Arctic seawaters (Dainard et al., 2015; Gonçalves-Araujo et al., 2016; Gao and Guéguen, 2017; Osburn et al., 2017; Chen et al., 2018; Brogi et al., 2019; D'Andrilli and McConnell, 2021)." (page 12, line 367−369).

In section 3.4. The authors could include in the supplementary material the Split-half graphs resulting from the validation of the components of the PARAFAC models.
(Response) We greatly appreciate Referee #2's insightful comment. As suggested, the Split-half graphs have been added to this revised manuscript. Please see the revised Figure 6 (page 37).

[Figure]

Figure 6: (a and b) Fluorescence EEM contour plots of the two fluorescent components C1 and C2 identified using EEM−PARAFAC in the fine-mode aerosols collected over the western Arctic Ocean during the summer of 2016. (c and d) The loading plots of C1 and C2 showing the split-half graphs.

In the conclusions section. The authors could emphasize how their study contributes to the assignment of emission sources of the WSOC present in the PM using the EEMPARAFAC tool.

(Response) We greatly appreciate Referee #2's insightful comment. According to Referee #2's insightful comment, we have improved the manuscript by adding the following descriptions to the Conclusions.

[revised manuscript text omitted]

---

## Author Response (AR2)

**Responses to the Editor's comments**

21/03/2023

Journal: *Atmospheric Chemistry and Physics*

Title: Measurement Report: Summertime fluorescence characteristics of atmospheric water-soluble organic carbon in the marine boundary layer of the western Arctic Ocean

Authors: Jinyoung Jung, Yuzo Miyazaki, Jin Hur, Yun Kyung Lee, Mi Hae Jeon, Youngju Lee, Kyoung-Ho Cho, Hyun Young Chung, Kitae Kim, Jung-Ok Choi, Catherine Lalande, Joo-Hong Kim, Taejin Choi, Young Jun Yoon, Eun Jin Yang, and Sung-Ho Kang

Manuscript number: acp-2022-665

Prof. Alex Huffman
Editor
Atmospheric Chemistry and Physics

Dear Professor Huffman,

We express our gratitude for allowing us to submit a revised version of our manuscript for publication in *Atmospheric Chemistry and Physics*. Your dedication and efforts in providing feedback on our paper are deeply appreciated. We are grateful for your insightful comments and valuable contributions, which have significantly improved the quality of our work. We have thoroughly incorporated all the suggestions that you provided.

We eagerly await your response regarding our submission and remain available to address any further queries or comments you may have.
Thank you for your consideration.

Sincerely,
Jinyoung Jung

Here is a point-by-point response to the editor's comments and concerns.
Note: The editor's comments are highlighted in **black**, while our responses to the editor are in **blue**. All modifications in the manuscript have been highlighted in yellow.

**Editorial comments:**

Thank you for your manuscript submission to Atmospheric Chemistry and Physics and for your detailed point-by-point responses to the two sets of referee comments. I think the data is important and can be useful for the community. I also think that the revisions you have processed for this version have improved the clarity of the presentation and I appreciate the addition of some statements about broader implications. As originally mentioned by Referee #1, however, I still feel that the scope of those broader atmospheric implications is still not very broad. While I think the manuscript is worth publishing, I think the it would be better suited to the journal as a measurement report. For this change, you only need to re-submit a newly revised manuscript with the words "Measurement Report: " appended to the beginning of your title. After doing so, I will quickly review to verify the change and will then accept the manuscript for publication.

(Response) We greatly appreciate the editor's careful review and thoughtful comments. According to the editor's insightful comment, we have revised the title of our manuscript to "Measurement Report: Summertime fluorescence characteristics of atmospheric water-soluble organic carbon in the marine boundary layer of the western Arctic Ocean" (page 1, line 1).

Additionally, I noted below a few grammatical points in the abstract that I suggest changing for better clarity:

L17: Move "in the rapidly changing Arctic Ocean" to before "including its summer time fluorescence characteristics" if you mean to imply the rapid changes are related to the WSOC and not just the fluorescence components

(Response) We greatly appreciate the editor's insightful comment. According to the editor's comment, we have relocated the phrase "in the rapidly changing Arctic Ocean" to before "including its fluorescence characteristics" (page 1, line 16).

L22: change to "high rates of sea-air gas exchange"

(Response) We greatly appreciate the editor's insightful comment. According to the editor's comment, we have revised "high sea–air gas exchange" to "high rates of sea–air gas exchange" (page 1, line 23).

L23: "increasingly available"

(Response) We greatly appreciate the editor's insightful comment. According to the editor's comment, we have revised "increasing available" to "increasingly available" (page 1, line 24).